# Reverse engineering model structures for soil and ecosystem respiration: the potential of gene expression programming

Iulia Ilie[1], Peter Dittrich[2,3], Nuno Carvalhais[1,4], Martin Jung[1], Andreas Heinemeyer[5], Mirco Migliavacca[1], James I.L. Morison[8], Sebastian Sippel[1], Jens-Arne Subke[6], Matthew Wilkinson[8], and Miguel D. Mahecha[1,3,7]

[1]Max Planck Institute for Biogeochemistry, Department Biogeochemical Integration, Hans-Knoell-Str. 10, 07745 Jena, Germany

[2]Bio Systems Analysis Group, Institute of Computer Science, Jena Centre for Bioinformatics and Friedrich Schiller University, 07745 Jena, Germany

[3]Michael Stifel Center Jena for Data-Driven and Simulation Science, 07745 Jena, Germany

[4]CENSE, Departamento de Ciências e Engenharia do Ambiente, Faculdade de Ciências e Tecnologia, Universidade NOVA de Lisboa, Caparica, Portugal.

[5]Department of Environment, Stockholm Environment Institute, University of York, York YO105NG, UK

[6]Biological and Environmental Sciences, School of Natural Sciences, University of Stirling, Stirling, UK

[7]German Centre for Integrative Biodiversity Research (iDiv), Deutscher Platz 5e, 04103 Leipzig, Germany

[8]Forest Research, Alice Holt Lodge, Farnham, Surrey, GU10 4LH, UK

*Correspondence to:* Miguel D. Mahecha (mmahecha@bgc-jena.mpg.de)

**Abstract.**

Accurate model representation of land-atmosphere carbon fluxes is essential for climate projections. However, the exact responses of carbon cycle processes to climatic drivers often remain uncertain. Presently, knowledge derived from experiments, complemented with a steadily evolving body of mechanistic theory provides the main basis for developing such models. The strongly increasing availability of measurements may facilitate new ways of identifying suitable model structures using machine learning. Here, we explore the potential of gene expression programming (GEP) to derive relevant model formulations based solely on the signals present in data by automatically applying various mathematical transformations to potential predictors and repeatedly evolving the resulting model structures. In contrast to most other machine learning regression techniques, the GEP approach generates "readable" models that allow for prediction and possibly for interpretation. Our study is based on two cases: artificially generated data and real observations. Simulations based on artificial data show that GEP is successful in identifying prescribed functions with the prediction capacity of the models comparable to four state-of-the-art machine learning methods (Random Forests, Support Vector Machines, Artificial Neural Networks, and Kernel Ridge Regressions). Based on real observations we explore the responses of the different components of terrestrial respiration at an oak forest in south-east England. We find that the GEP retrieved models are often better in prediction than some established respiration models. Based on their structures, we find previously unconsidered exponential dependencies of respiration on seasonal ecosystem carbon assimilation and water dynamics. We noticed that the GEP models are only partly portable across respiration components; the identification of a "general" terrestrial respiration model possibly prevented by equifinality issues. Overall, GEP is a promising

tool for uncovering new model structures for terrestrial ecology in the data rich era, complementing more traditional modelling approaches.

**Highlights**

– We explore if the process of model building for describing ecosystem $CO_2$ fluxes can be, to a large extent, automated .

– We show that Gene Expression Programming combined with parameter optimization can be a useful algorithm to automatically derive models from ecological time series.

– We propose alternative models for the influence of key environmental variables on various respiratory fluxes $CO_2$ in an oak forest.

– Conventional ecosystem response functions can be revised by new models identified with gene expression programming.

# 1 Introduction

One prerequisite to understand and anticipate the global consequences of anthropogenic climate change is an accurate quantitative description of the terrestrial carbon cycle (Bonan, 2008; Heimann and Reichstein, 2008; Luo et al., 2015). However, the description of the mechanisms underlying the total terrestrial efflux of $CO_2$ (Peng et al., 2014a), often referred to as "terrestrial ecosystem respiration" ($R_{eco}$), varies across the scientific literature and existing global models. This is partly because $R_{eco}$ does not originate from a single process but is the sum of fluxes from different autotrophic and heterotrophic respiration processes that operate across different temporal and spatial scales and compartments (e.g. soil depths). Hence, it is experimentally very difficult to disentangle the main abiotic and biotic factors driving respiratory processes at the ecosystem level (Trumbore, 2006) and to derive suitable models for the individual respiration processes. In the remaining manuscript we use the term "model" as an equivalent of "response functions" i.e. some analytic description of how environmental drivers influence ecosystem fluxes.

Traditionally, respiration models have been based on some theoretical considerations but largely remain empirical in nature (e.g. Reichstein and Beer, 2008; Gilmanov et al., 2010; Hoffmann et al., 2015). Conventional model building (Fig. 1) is primarily hypothesis driven and capitalizes both on some understanding of the system and reported scaled experiments (Migliavacca et al., 2012; Richardson et al., 2008). Gupta et al. (2012) describe this common paradigm of model development as a four step approach involving  a) observational, b) conceptual, c) mathematical and, d) computational phases (see also e.g. Bennett et al., 2010; Williams et al., 2009). During the observational phase, the system under scrutiny is monitored and observations are assembled, ideally representing process responses to hypothesized driving variables. Based on these observations, a conceptual model is proposed, which is subsequently guiding the formulations of mathematical representations of the system states and dependencies. The mathematical description then provides the basis for computational models that are used for simulations (Jakeman et al., 2006). Model-data integration may additionally lead to iterative structural revisions or parameter optimizations

(Williams et al., 2009). This conventional approach to model development is also characteristic of different kinds of ecological model building, including the development of biogeochemical models (Williams et al., 2009).

We explore the possibility of reverse engineering offering an automated alternative to model development for predicting terrestrial carbon fluxes (Fig. 1). In reverse engineering, the work flow is fundamentally different (Bongard and Lipson, 2007):

*a*) database set-up phase, *b*) computational phase, *c*) mathematical phase and a *d*) conceptual phase (Gupta et al., 2012). The rationale behind reordering the key phases is firstly to minimize the human influence and perception biases that might shape the formulation of new hypotheses, and secondly to increase the chance for novel model structures to automatically emerge from the available data and that would not be so obvious from a direct analysis. Reverse engineering is aiming at identifying some mathematical representation of a system that is to a large degree independent from a priori conceptualizations; in the

current case, the respiratory response of terrestrial ecosystems to environmental drivers. Reverse engineering leaves the model construction up to an algorithm and is therefore a way to empirically learn from observations with minimal user input.

Of course, expert knowledge still has a large influence on the modelling process, as only a certain set of variables can be measured and even a smaller subset is indeed available for model development, which includes the restriction to a certain plausible number of time lags, and hence full objectivity of automatic model development cannot be truly achieved. Further-

more, expert knowledge comes into play when the algorithm is set for running, by tuning the set of parameters according to the problem needed to be solved and as well during the observation collection and during the final decision on whether the solution returned by the algorithm actually makes sense at all and whether it can be further used. Nevertheless, we believe that by shifting the moment when the analyst make the decision regarding the selected model, a larger degree of objectivity in modelling is achieved.

Reverse engineering is close to machine learning based regression techniques, where various candidate model formulations and specifications are explored in order to minimize the prediction error. The fundamental difference from typical model building is that reverse engineering typically provides a symbolic regression, that is, the resulting structures are ideally directly readable as mathematical functions (i.e. response functions) and can be interpreted. The readable character of the returned solutions allows to consider the applicability of the derived structures in other system domains (Ashworth et al., 2012).

Here, we focus on the "Gene Expression Programming" (GEP, Ferreira, 2001) reverse engineering approach. GEP is an evolutionary algorithm that constructs mathematical response functions. In its essence, GEP basically converges to a solution after rejecting a large number of potential regression models over a certain amount of evolutionary steps. Due to its structural design, GEP can be applied in a wide range of empirical modelling problems (Peng et al., 2014b; Khatibi et al., 2013; Traore and Guven, 2013), including (soil) hydrology (Fernando et al., 2009; Hashmi and Shamseldin, 2014). To the best of our

knowledge the potential of GEP has not yet been explored for modelling biogeochemical fluxes in terrestrial ecosystems.

We seek to understand as well whether automating model development can provide new insights in understanding the dynamics of terrestrial respiration processes. We base our study on data from a long-term monitoring experiment of $R_{eco}$ components i.e. above ground respiration, root respiration, mycorrhiza respiration, soil autotrophic, and soil heterotrophic respiration. The monitoring was done separately but in a time-synchronized way over two years and is described in detail by

Heinemeyer et al. (2012).

The fundamental question addressed in this paper is whether regression models can be constructed more objectively by leaving the task of proposing a final regression model to an algorithm rather than directly to an analyst. The need for human intuition during the actual process of constructing a regression model becomes reduced, and the input of expert knowledge shifts towards identifying input variables, parameters, a suitable cost function and model plausibility.

With the current study we investigate as well if automatically derived model structures differ substantially from models conventionally used in the study of $R_{eco}$ and its components or, if they are consistent with established theory. The separation of $R_{eco}$ into its components also allowed us to test the portability of individual model structures across different respiration components. In this sense, we investigate whether a generic "respiration" response can be derived, or if specific formulations for a range of respiration components are required.

## 1.1 Study structure

First, we introduce the GEP methodology and explore its performance for symbolic regression type of problems using an artificial experiment under varying degrees of noise contamination designed to resemble $R_{eco}$. Second, we apply GEP to model the various respiration observations provided by Heinemeyer et al. (2012).

The observational record provided by Heinemeyer et al. (2012) is exceptional, because measurements of soil or ecosystem respiration that are typically only integrated, are here continuously and regularly measured, and the components measured offer a perfect test case for the GEP methodology.

For both the artificial experiment and real world observations, we systematically confront the prediction error of GEP with other state-of-the-art machine learning regression approaches. In addition, we adjust the modelling approach such that the objective function (or fitness function) accounts not only for absolute or relative error, but also reduces structure in the residuals. The discussion focuses on the comparison of the various GEP derived models, their equifinality, and performance compared to widely used literature models.

## 2 Method

We rely on the GEP method (Ferreira, 2001) which automatically constructs model structures based on a set of given observations. As the models we want to obtain are mathematical structures, their construction can be achieved by solving a symbolic regression (Kotanchek et al., 2013) type of problem. That is, we are not only interested in determining an optimal set of parameters for a known regression, but here, we want to discover the symbolic form of the regression itself by identifying the most important predictors and their functional transformations. The general GEP approach in solving symbolic regressions is presented in the following section and is illustrated in Fig. 2.

### 2.1 Gene Expression Programming, GEP

The process of finding the most suitable model structure based on signal present in data in GEP starts with an initial generation of $n$ possible model structures (Fig. 3, A). These can be called evolution individuals and in GEP, they are known

as "chromosomes". The chromosomes are composed of a fixed number of "genes" that are connected by a binary mathematical operator. Each gene is encoded in a string with a fixed length that contains specific characters that map to either a set of possible predictors, e.g. $A = \{a, b\} \rightarrow A_m = \{x_1, x_2\}$ or a set of their possible functional transformations, e.g. $F = \{+, -, L, E\} \rightarrow F_m = \{\text{addition,substraction,logarithm,exponential}\}$, (see Fig. 3, A).

The choice of input functions used for applying mathematical transformations on the predictors depends on the type of problem we try to solve with GEP. When the problem is a symbolic regression type of problem, as here, most often a set of primitive functions is proposed; such as addition, multiplication, exponential and so on. More complex functions could increase model complexity too much and risk over fitting. However if there are already known functional transformations of certain predictors that could be part of the final desired solution, the user can define a new function and introduce it in the set

of input functions.

     All genes are made up of a " gene head", containing a combination of characters mapping to both predictors and functional transformations and a " gene tail", with characters that map only to predictors. The gene length is given by $g_l = h_l + t_l$, where $t_l = (f_{max} - 1) \times h_l + 1$, with $g_l$ as gene length, $h_l$ head length, $t_l$ tail length and $f_{max}$ as the maximum parity of a functional transformation.

As in biology evolution, regardless of the actual length, the GEP genes have active sections of variable length called "open reading frames" (ORF) that can encode various expression trees which can be evaluated into mathematical expressions (Ferreira, 2006). The lengths of the ORFs are determined only after the encoded expression trees are translated using an internal reading language (see Fig. Fig. 3, B). Ferreira (2001) argues that, the power of GEP lies in its use of fixed length linear strings for representing expression trees (ET) of varied shapes and sizes that simplifies the evolutionary process, and helps reach a

final solution faster.

     The total number of chromosomes generated over each evolution step make up the GEP population. The evolution steps are also known as "generations". The maximum number of generations allowed to run until reaching a solution is often used as a stopping criterion.

     One of the crucial components of model developing within an evolutionary algorithm is the selection process. In GEP,

the chromosomes can be translated into mathematical expressions that can be evaluated, and a distance between the current structure based predictions and the original target is computed. The measures are known as "fitness values" and are assigned to all the chromosomes in the population at each generation by means of a predefined fitness function. The evolution of the final solution with GEP is done based on optimizing the fitness function values after each generation, usually by minimizing prediction error, but more complex criteria can be taken into account as well.

Once all the fitness values have been computed and assigned, the chromosomes in a generation are sorted from best to worst fit.

     If no stop criteria has been met, preparations for the reproduction of new chromosomes for the next generation are made. The chromosome with the best fitness value is reproduced unchanged in the first position of the new generation. For filling the remaining n-1 positions, chromosomes are selected from the entire population for the new generation with a tournament

procedure, n-1 times.

In tournament selection, 2 chromosomes are randomly selected from the entire population and the individual with the better fitness value goes through.

For insuring that novel material is introduced in the pool of possible model structures, n-1 newly selected chromosomes are subject to genetic operators, such as: mutation, recombination, transposition and inversion as presented in Fig. 3, D, that can fully change the encoded mathematical expressions (see Fig. 3, C).

Once the population of chromosomes is ready for the new generation, the evolution procedure is repeated until a stop criterion is reached, such as best fitness achieved, maximum number of unimproved generations is reached, time limit, etc.

The hyper-parameter needed for a GEP run, i.e the set of all parameters that need to be fixed before a GEP run is performed, has either components with recommended default values, especially for the genetic operator rates considered when applying the available genetic operators (Ferreira, 2006), or has components for which the values have been established empirically after experience in working with the GEP approach. The latter typically depend on the requirements of the problem looked to solve.

Such is the case for setting the length the gene head, or the number of genes in a chromosome that can be lower if the interest is in obtaining more compact solutions, with larger values possibly leading to a fast expansion of solution length which can easily over-fit the initial target. When the lengths of the chromosomes are kept too low, the structures in the population can convergence too soon to a unique solution that might lack the ability to capture meaningful signals present in the training data, due to low diversity of the encoded expression trees.

Another important component of the hyper-parameter to fix is the mutation rate which is one of the genetic variation operators. When the mutation rate is too large, it can become disruptive and lead to loss of information acquired along the previous evolutionary time steps, reducing the general convergence of the GEP run. Conversely, if the rate is too low, relevant structures may not be constructed in the given time limit.

The current implementation of the GEP approach does not contain an explicit population diversity management component which could increase the confidence that a certain solution did not just appear by chance, but that it was actually selected over a larger pool of possible model structure types. In order to reduce stochastic bias and avoid getting stuck in local optima that would produce over-fitted results, we chose the practical approach of multi-start (multiple runs with the same settings) as proposed by Ferreira (2006).

The version of the GEP method presented in this paper was implemented by the first author in the C++ language and is freely available upon request. All the experiments reported in this work were executed on a cluster running SuSE SLES 11 SP1 and StorNEXT (global file system running on the IO nodes) and that contains 868 CPU cores, 14.5 TB RAM, 1.2 PB file space. The large performance capacity of the cluster allowed for multiple parallel runs and speed in reaching the final solutions.

## 2.2 Fitness measure

In our study, the fitness measure is reported in terms of the Nash–Sutcliffe modelling efficiency (MEF) coefficient (Nash and Sutcliffe, 1970; Bennett et al., 2010) which is often used in the context of quantifying the performance of terrestrial biosphere

models (Mitchell et al., 2009; Migliavacca et al., 2015). The MEF is computed as

$$
\text{MEF} = 1 - \frac{\sum\limits_{i=1}^{n}(o_i - p_i)^2}{\sum\limits_{i=1}^{n}(o_i - \bar{o})^2} \tag{1}
$$

where $o_i$ is the observed value at step $i$ and $p_i$ is the predicted value at step $i$ and $\bar{o}$ is the mean of observed values. MEF values range between $-\infty$ and 1, where an MEF value of 1 corresponds to the case where the predicted and observed values are identical. A negative MEF value means that the predictions are worse than the mean of the observations in recreating the observed signal. MEF=0 indicates that the models prediction are as good as a prediction by $\bar{o}$.

During the GEP learning process, however we use the (1-MEF) measure as we want to minimize the fitness function values.

Although the MEF metric offers a straightforward interpretation, it does not take the number of parameters of the models into account. In real-world applications, it might be desirable to derive models with fewer parameters if those are not (much) worse in terms of prediction capacity than models with higher number of free terms. Thus, we include in our cost (fitness) function a normalized term related to number of parameters (ratio of current number of parameters to maximum number of possible parameters given the GEP run settings).

Moreover, any systematic pattern in the model residuals needs to be reduced as the latter should ideally only represent uncorrelated noise. To meet this criterion, we complement the fitness function with a term related to the information content (entropy) in the residual time series. Entropy values would be maximized for data without structure (i.e. white noise), and lower entropy values would be obtained for structured data, e.g. correlated stochastic or deterministic processes (Rosso et al., 2007) . The information content in a time series is typically quantified by the Shannon Entropy (SE, C. E. Shannon (1948)) , i.e. a term of the form

$$
\text{S}E(X) \; = \; -\sum_{i=1}^{N} p_i \ln [p_i] \; . \tag{2}
$$

Here, $X = \{p_i; i = 1, \dots, N\}$ denotes a probability distribution with $\sum_{i=1}^{N} p_i = 1$ and $N$ possible states.

In short, the calculation of an entropy as a measure for randomness from a time series (e.g. Shannon's entropy) requires to determine a probability distribution that underlies the time series (or dynamical system), which is usually done by a partitioning step (also called phase space reconstruction in other contexts). This is a fundamental step in the methodology, and various methods have been used to arrive at this probability distribution, for instance frequency or histogram-based measures, procedures based on amplitude statistics, or symbolic dynamics (see e.g Kowalski et al. (2011) for an overview).

As our aim is to minimize structure in the residuals, the temporal order becomes important. In recent years, the Bandt-Pompe approach has become popular, because it directly takes time sequences into account: The technique hence divides the time series into ordinal sequences (i.e. ordinal patterns, or symbolic sequences), and then computes entropy measures directly from the probability distribution of these ordinal patterns (Bandt and Pompe, 2002).

This approach has a number of advantages, namely that it is robust to noise (no sensitivity to numeric outliers) and to trends or drift in the data, it is an (almost) non-parametric method and no prior assumptions about the data are needed (the only

parameter that has to be specified is the embedding dimension, i.e. window length), and allows to disentangle various possible states of the system that are then encoded in the probability distribution (see e.g. Zanin et al. (2012) for a review of the method and applications).

The single parameter that needs specification is the window length. This parameter is fixed to $n_{demb} = 4$ throughout the entire manuscript following previous work on ecosystem gross primary productivity dynamics by Sippel et al. (2016).

5    The final normalized form of the fitness function further used in our work is:

$$\text{CEM} = \sqrt{(1 - \text{MEF})^2 + (\frac{P}{P_{max}})^2 + (1 - \text{SE})^2} \tag{3}$$

$$P_{max} = n_g \times l \tag{4}$$

where, CEM stands from here on for "complexity corrected efficiency in modelling", $P$ is the number of parameters present in a model structure, $P_{max}$ is the maximum number of parameters possible for each individual from a GEP run set-up, $n_g$ is the number of genes in a chromosome and $l$ is the length of a gene.

For assessing the effect of adding the entropy component for the residuals in the CEM fitness function, we introduce as well a fitness measure containing elements regarding only the MEF and the number of parameters.

$$\text{MEF+NP} = \sqrt{(1 - \text{MEF})^2 + (\frac{P}{P_{max}})^2} \tag{5}$$

For all experiments reported in this paper, the optimization is done by minimizing the CEM fitness function values. The best value that can be reached for all presented fitness functions is 0.

## 2.3    Parameter optimization

The GEP algorithm does not have a specific treatment of constants in the building of model formulations but mutations can change both the model structure and constants. However, the scaling of constant values (model parameters) might be a decisive
factor in adequately determining the fitness of a formulation. Without this, a model structure might be discarded regardless of potentially being a very powerful candidate. Furthermore, model parameters are often very informative regarding a system's sensitivity to some modifications of the drivers. These aspects have led to the addition of a final parameter optimization step at the end of each GEP run.

In order to obtain an optimal set of parameters for the GEP extracted model structures, an approach that would be applicable
in a large set of generated search spaces was necessary. Here we use the "Covariance Matrix Adaptation Evolution Strategy" (CMA-ES, Hansen et al. (2003)) for optimization. The CMA-ES is a stochastic optimization algorithm that seeks to minimize a fitness function by estimating and adapting a covariance matrix according to a sampling from a multivariate normal distribution (Beyer and Schwefel, 2002; Auger and Hansen, 2005). According to Hansen (2006), one of the main arguments in favour of the CMA-ES approach is that it has shown good results even in the case of ill-posed problems (Kabanikhin, 2008), which may very well be the case for some of the GEP structures that are automatically generated.

The CMA-ES version used for the final step of optimization is the Hansen Python implementation found at https://pypi.python.org/pypi/cma.

## 3  Experimental design

For exploring the possibility of using GEP in developing relevant model structures for describing the terrestrial carbon fluxes, two case studies were designed: Firstly, an experiment based on artificially generated data to better understand and present the general properties and capacities of GEP. Secondly, we explored the use of GEP on real measurements of various respiratory flux components monitored continuously over two years in an oak forest (Heinemeyer et al., 2011).

### 3.1  Artificial experiments

These experiments were designed to explore whether our implementation of the GEP method is suitable for symbolic regression type of problems, and how robust/vulnerable it is across various signal to noise ratios. We explored a set of functions with increasing levels of non-linearity to generate data points.

$$f(x_1) = 2x_1 + 1 \tag{6}$$

$$f(x_1) = x_1^2 + 3x_1 + 5 \tag{7}$$

$$f(x_1) = e^{x_1} + 1 \tag{8}$$

$$f(x_1) = e^{-x_1} - x_1 \tag{9}$$

$$f(x_1) = x_1^2 - 4\sin(x_1) \tag{10}$$

$$f(x_1) = x_1^3 + 6x_1^2 + 11x_1 - 6 \tag{11}$$

$$f(x_1, x_2) = x_2 x_1 \tag{12}$$

$$f(x_1, x_2) = x_2 x_1 - 3\cos(x_1) \tag{13}$$

$$f(x_1, x_2) = 2x_1^2 + 3x_2^2 \tag{14}$$

$$f(x_1, x_2, x_3) = 2x_1^2 + 3x_2^2 + 2\sin(x_3) \tag{15}$$

2000 data points were randomly generated with $x_1 \in [1, 20]; x_2 \in [1, 5]; x_3 \in [1, 100]$ and all the functional transformations were done based on the same initial set of 2000 data points. Out of the 2000 data points, 1000 data points were used for training, while 1000 data points were reserved for validation. The GEP settings used for each of the 20 runs are given in Table 1. If a returned structure was identical to the originally prescribed function or if $(1 - \text{MEF}) \leq 10^{-5}$ at validation, the retrieval of the original structure was considered to be a success. For allowing the approaches to do an automatic feature selection, all 3 variables, $x_1, x_2, x_3$, were used for learning and validation for all 10 functions in the benchmark set.

For investigating the capacity of GEP to reconstruct a simple model used in the ecology field as well, we introduced as well an artificial test for the "$Q_{10}$" model that is used in the field for simulating the response of ecosystem respiration to change in

air temperature of $10^oC$ at a reference temperature of $15^oC$ The formulation we used for the "$Q_{10}$" model is:

$$R_{eco} = 2^{(0.1T_{air}-1.5)} \tag{16}$$

with $R_{eco}$ as ecosystem respiration flux and $T_{air}$, the air temperature. Again, we generated 2000 data points for both predictor
and target and we used half for training 100 runs and half for validation. The modelling capacity of the best structure in terms of fitness value at validation is reported.

In order to investigate the response of the GEP approach to noise contaminated data, we simulated Gaussian noise that scales with signal amplitude as often observed in the case of terrestrial ecosystem (Lasslop et al., 2012) and soil respiration (Lavoie et al., 2015)fluxes. The signal-to-noise ratio (SNR, measured as ratios of standard deviations) was varied between 10 and 1 in
six steps.

For each of these functions and SNR levels, we sampled 100 validation data points 10 times. 20 GEP runs were performed on the 1000 training data points and the GEP model structure with the highest mean MEF value over the 10 validation sets was chosen.

As the choice of fitness function was crucial for the construction of structures in a GEP type of approach, we also investigated
in one experiment the effects of minimizing the CEM values (eq. 3) as opposed to using only MEF (eq. 1) or MEF+NP (eq. 5) as fitness function.

### 3.1.1  Alternative Machine Learning Methods

The prediction performance of the best GEP derived models based on the data in section 3.1 was compared with the prediction performance of four commonly used state-of-the-art machine learning methods (MLM), i.e Artificial Neural Networks, ANN,
(Yegnanarayana, 2009), Support vector Machines, SVM (Hearst, 1998), Random Forests, RF (Breiman, 2001) and Kernel Ridge Regressions, KRR (Hoerl and Kennard, 1970).

The toolboxes and settings used for generating the predictions by the ANN and KRR methods are described by Tramontana et al. (2016) and found in the "simple R" regression toolbox (Lazaro-Gredilla et al., 2014). The predictions of the SVM were obtained by using the "LIBSVM" library (Chang and Lin, 2011) from the "SimpleR" regression toolbox where the regulariza-
25 tion term, the insensitivity tube (tolerated error) and a kernel length scale were automatically adjusted during each run. Lastly, the RF predictions were obtained after running the MATLAB statistics toolbox implementation with default settings. The hyper-parameters of all MLM were estimated to avoid over-fitting during each run as presented in section S6 of Tramontana et al. (2016).

All the present machine learning approaches have been applied on the same training data sets as those used for building the
30 GEP models, and their predicted values were compared with the validation sets used for determining the best GEP solution.

### 3.2  Measured ecosystem $CO_2$ fluxes

In the second experiment we assessed the possibility to reverse engineer model structures $R_{eco}$ and its components based only on real measured data. Specifically, we explored GEP derived model structures for various components of terrestrial ecosystem

respiration fluxes measured in an 80 year old deciduous oak plantation in the Alice Holt forest in SE England as described in (Heinemeyer et al., 2012; Wilkinson et al., 2012).

### 3.2.1 Alice Holt in-situ data

The Alice Holt data set contains observations of $R_{eco}$ and the total influx of $CO_2$ to the ecosystem as mediated via photosynthesis (gross primary production, $GPP$), and various soil respiration components.

$R_{eco}$ and $GPP$ were estimated from eddy covariance measurements of the forest net $CO_2$ exchange (NEE, Eq. 17) and were obtained from a micro-meteorological measurement tower at the same site that reports half hourly integrals of NEE with the

10 eddy covariance (EC) methodology (Moncrieff et al., 1997). The Reichstein et al. (2005) procedure was used for gap-filling and separation of NEE into $GPP$ and $R_{eco}$. Given that $R_{soil}$ is a fraction of $R_{eco}$, above ground respiration can be calculated as the difference between $R_{eco}$ and $R_{soil}$. For an in-depth description of other site conditions and measurements see Heinemeyer et al. (2012).

A multiplexed chamber system was used for separately measuring soil respiration ($R_{soil}$ ) and its components, using a

15 continuous sampling method at fixed locations during two years at an hourly resolution. In order to partition the $R_{soil}$ flux into its components, mesh-bags that are not penetrable by roots, but allow for mycorrhizal hyphae development were installed. Deep steel collars were applied to stop both root and mycorrhizae in-growth. As a result, root respiration ($R_{root}$) is given by the difference of $R_{soil}$ and the respiration recorded in the mesh bag chambers, mycorrhiza respiration ($R_{myc}$) is given by subtracting the steel collar flux from the mesh bag chamber flux, and the soil heterotrophic respiration ($R_{soil_h}$) is given by the

20 $CO_2$ efflux at the steel collar chambers. Lastly, soil autotrophic respiration ($R_{soil_a}$) is estimated as the sum of $R_{myc}$ and $R_{root}$ (Eq. 19 and 20) .

The above ground respiration ($R_{above}$) was given as well and was estimated by difference (Eq. 18). Additionally, direct measurements of soil moisture ($SWC$), air temperature, surface temperature, and soil temperature taken at 2, 10 and 20 cm depth are present in the dataset.

$$R_{eco} = \text{NEE} + GPP \tag{17}$$

$$R_{above} = R_{eco} - R_{soil} \tag{18}$$

$$R_{soil_a} = R_{root} + R_{myc} \tag{19}$$

$$R_{soil} = R_{soil_a} + R_{soil_h} \tag{20}$$

The computation of $R_{above}$ as difference between $R_{eco}$ and $R_{soil}$ might be highly uncertain because of the different tech-

30 niques used to compute the two respiration components, the completely different footprints, and the typical high flux underestimation and low flux overestimation of $R_{eco}$ from EC (Wehr et al., 2016). The limitations of the separation of $R_{eco}$ into its components and the uncertainty of the estimates are further discussed by Heinemeyer et al. (2011), Heinemeyer et al. (2012) and Wilkinson et al. (2012).

### 3.2.2 Data processing

We used the following candidate driver variables: soil volumetric moisture measurements, air temperature (from micro-meteorological station), and temperatures at different soil depths, and $GPP$. A number of recent studies have shown a tight

linkage between $GPP$ and $R_{soil}$, reflecting dynamics of respiratory substrate supply to roots and mycorrhizal fungi from recently assimilated C in plants. (Moyano et al., 2008; Mahecha et al., 2010; Migliavacca et al., 2011, amongst others). We use $GPP$ obtained from EC measurements at the site, but acknowledge the conceptual problem that $R_{eco}$ and $GPP$ were derived from the same observations of NEE. In order to minimize the potential spurious correlation between $R_{eco}$ and $GPP$ as well as redundancy of possible $GPP$ influence with the meteorological drivers, we considered low-frequency variability of $GPP$

only (i.e. low-pass filtered modes of $GPP$ which corresponds to variability beyond a 60 days periodicity only, see Mahecha et al., 2010). "Singular Spectrum Analysis" (SSA, Broomhead and King (1986)) as described and implemented by Buttlar et al. (2014) was used to obtain a smooth $GPP$ signal. The seasonal cycle was extracted with the SSA method as the assumption is that $GPP$ affects mainly the seasonality of the respiration while the variability at the high frequency is assumed to be more related to meteorological drivers (e.g. temperature, Mahecha et al., 2010). The SSA method is a tool used mainly in time series

analysis with the purpose of decomposing a time series signal into its independent sum components, such as trends, seasonality and high frequency components based on a singular value decomposition of trajectory matrices computed after embedding the time series (Buttlar et al., 2014).

To reduce the skewness and the search space that the GEP evolution would have to cover in order to construct valuable solutions (Keene, 1995), we log-transformed the seven target respiration data sets (see Figure 1 in supplemental material) and

20 applied a back-transformation when reporting the respective model structures. Manning (1998) and Newman (1993) show that when regressions are built based on log transformed targets, the back-transformation of the regressions to non-transformed target needs to include a bias correction that refers to the residuals of the log models.

As such, if the log model is $logy = \alpha x + \epsilon$, the back transformation to $y$ should not simply be $y = e^{(}\alpha x)$, but should include a correction of the bias induced by $\epsilon$, and depending on the distribution of the residuals, the back-transformation can be:

$y = e^{(\alpha x + 0.5\sigma_\epsilon^2)}$, when the residuals are log normal distributed;

     $y = e^{(\alpha x)} E(e^\epsilon)$, where E is the mean of the sample, when the residuals show heteroscedasticity, as was the case for most of the residuals computed for the GEP models as seen in Fig. 2 of suppl.;

     $y = e^{(\alpha x)}$ if no bias correction is desired, or a naive approach.

The time series used for the candidate drivers observations remain unchanged.

### 3.2.3 GEP set-up

For each combination of respiration target and possible drivers, 50 subsets of 500 target time steps each, were randomly selected and used for the training of GEP models using the settings found in Table 1. The 50 subsets of the remaining 113 time steps are

used for cross-validation and the model with the lowest average validation CEM value is finally selected for each respiration type. For all runs the observations are given as records of daily mean values.

We were particularly interested in determining the general character of each extracted model with respect to the different respiration fractions. We therefore re-optimized the parameters of all extracted model structures when applying one extracted model as the candidate function for a different respiration term. For example, the model formulation extracted for $R_{eco}$ is re-calibrated for all the other types of respiration, creating six parameter sets (one for each respiratory flux) per equation. To cross-validate parameter sets, we computed performances for each train–validation data set pair and report averaged MEF values.

As in the artificial example, we compared the returned GEP solutions predictions performance with that of other common MLM such as SVN, KRR, ANNs, and RF. All methods were used for generating 50 subsets of 113 prediction values, after training on the 50 subsets of 500 time steps of observations presented in the start of section 3.2.3. Then, a mean MEF value was computed for all methods for all respiration components and the best mean MEF values were reported and compared with those of the GEP extracted models. The comparison is done in terms of MEF as number of model parameters were not available and CEM could not be computed.

### 3.2.4 GEP in the context of other known ecological models: Real observational data

A comparison was done between the GEP built models and some common literature respiration models with different structures and driving variables that were also optimized using CMA-ES. The optimization was performed for each respiration dataset and its candidate drivers and parameters (Table 2). The structures and prediction performances of the GEP models were then compared with those of the optimized literature models.

## 4 Results

### 4.1 Artificial experiments

In the first artificial experiment the GEP approach is used to verify if it can reconstruct prescribed functions. Following the training of the 20 independent GEP runs, the initial functions were successfully reconstructed for all 10 equations defined in section 3.1.

For the $Q_{10}$ model artificial test, the following structure was finally selected:

$$R_{eco} = 0.35 \times 2.5^{(0.01 T_{air})} \tag{21}$$

with a validation MEF value $> 0.99$.

MEF values for the GEP extracted models and for the predictions generated by ANN, RF, KRR and SVM are illustrated in Fig. 4. These MEF values were obtained through cross validation against independent, yet equally noise contaminated data points (the SNR values are given on the x axis in reverse order for visualizing the increase in noise levels). There is a clear

pattern of decreasing MEFs with increasing noise contamination. This was expected, as none of the methods should fit the
noise added to the signal.

Figure 4,B, shows MEF values equivalent to Fig. 4,A, but applied to noise-free data points of the validation set, in order to
compare GEP outputs to the "true" structure underlying the artificial data set. In this set-up, the MEF values remained relatively
constant across SNR values above 2. When SNR level was set to 1, predictions for all investigated machine learning methods,
except for GEP predictions, show decreased fitness, with MEF values decreasing to a minimum of 0.8.

In order to verify the effects of changing the fitness function from MEF to CEM, we compare the distributions of MEF values
for all runs for all studied SNR. Figure 5 exemplifies outputs for equation 15; panel a shows a drop of prediction capacity of
the GEP models with noise increase for all types of fitness functions when compared with noise-infused data. This contrasts
the reduced MEF assessed against original data, where a slight drop in MEF with noise increase for the MEF optimization
structures was seen, and where the CEM optimized structures show stability in MEF with noise. The new CEM leads to a
reduced number of returned parameters compared to MEF (Fig.5c), as well.

## 4.2 Measured ecosystem $CO_2$ fluxes

Applying GEP on the Alice Holt data set yielded a series of model structures for each respiration type. The returned model
structures after bias-corrected back-transformation are illustrated in equations 22-28.

$$R_{eco} = 1.2 \log \left( T_{-10} \right)^{0.8} \times e^{\left( \frac{GPP_s}{T_{-10}} \right)} \tag{22}$$

$$R_{above} = 1.1 SWC^{0.3} \times e^{(0.1 GPP_s)} \tag{23}$$

$$R_{soil} = 0.04 e^{(1.1 T_{-10}^{0.4} + 1.6 SWC)} \tag{24}$$

$$R_{root} = 1.1 e^{\frac{0.9 GPP_s - 6.8}{T_{-10}}} \tag{25}$$

$$R_{myc} = 0.001 T_{-10}^{1.2} \times e^{(1.6 T_{-10})^{SWC}} \tag{26}$$

$$R_{soil_a} = 0.01 e^{(0.8 T_{-10}^{0.6} + 2.6 SWC)} \tag{27}$$

$$R_{soil_h} = 0.8 e^{\frac{0.6 GPP_s - 2.4}{T_{-10}}} \tag{28}$$

where, $GPP_s$ is gross primary production that has been smoothed using the SSA method with a 60 day window ; $T_{-10}$
is soil temperature measured at 10 cm depth; and $SWC$ is volumetric soil water content. The corresponding cross-validation
MEF values are given in Table 3, indicating a range of capacities for GEP models to represent different respiration types.

Whilst GEP-derived models may differ between respiration types, there are a number of equivalent models for different
respiration components. $R_{soil}$ and $R_{soil_a}$ were described by identical model structures (but distinctive parameter values), and

$R_{root}$ and $R_{soil_h}$ were described by similar (but not identical) models. Overall, the most common selected drivers were $T_{-10}$, $SWC$ and $GPP$.

The highest performance in terms of MEF value was recorded for $R_{soil_a}$ and for $R_{soil}$, that is 0.82 and 0.81 respectively. The lowest capacity of process representation, with an MEF value of 0.28, was recorded for $R_{above}$ (Table 3), possibly because this specific component would need to include active versus inactive periods determined by dormancy and leaf fall (i.e. seasonality in this deciduous forest). A comparison of the predicted values and observed fluxes for all types of respiration can be seen in Figures 6 and 8. Figures 7 and 9 show the effects of the three different types of bias correction on the global signal reconstruction and prediction capacity with MEF values computed in a cross-validation manner. For all respiration types, except $R_{soil}$ and , doing the second type of bias correction, with a smear term improved the prediction capacity. Although for $R_{soil}$ it seems that doing no bias correction gives a higher MEF value, we chose to keep the model including the smear term.

In order to explore the capacity of the GEP models generated for the $R_{eco}$ components to recreate the larger, across compartmental summed fluxes, we summed the predictions of the models and compared them with the original fluxes (Fig. 10).Based on a modelling performance comparison of the models defined as sum models of the initial GEP models trained on the component fluxes with the original GEP models trained on the summed fluxes, we found no significant differences after performing Student's t-test (h=0, p=0.5). However, we found that the total number of parameters is much larger for the sum models. This can be a result of the GEP approach eliminating the "low impact " drivers due to complexity pressure. We can see as well that the sensitivity of the sum fluxes to certain drivers can strongly manifest itself only in certain components which is why the drivers only get selected in the models built for those specific components.

The residuals depict some remaining patterns (Fig. 11 and Fig. 3 of suppl.) and the null hypothesis of normal distribution was rejected for all seven respiration component residuals at 5% significance level with the one-sample Kolmogorov-Smirnov test. Hence, we might expect additional information that could be extracted from the residuals. In order to check whether the remaining structure was missed in the first training routine because of imposing a multiplicative form in the models by log-transforming the target data, we performed GEP runs on the residuals and combined the models. The improvement in overall modelling performance is minimal, yet model structures become overly complex. The capacity of the GEP approach to retrieve new information from the residuals is illustrated in Fig. 13 in comparison with that of the other MLM presented in section 3.1.1. When correlation values were computed between the candidate drivers and the residuals, no significant linear correlations were found (Fig. 5 and 6 of suppl.).

### 4.2.1 Model transferability

We investigated the capacity of each extracted model structure (equations 22-28) to represent a component of $R_{eco}$ not seen in the training procedure. This was done by means of new CMA-ES optimization steps. The new prediction performances are illustrated in Tab. 4.

After optimization, none of the structures show an overall best MEF for all the $R_{eco}$ components (i.e. we clearly cannot identify an optimal general model). However, we identify certain model structures that tend to perform overall better than

others. This is the case for the $R_{myc}$ model (eq. 26). It can also be seen that after the individual model optimizations, the structures for $R_{eco}$ and that for $R_{soil_a}$ have similar prediction capacities.

    The prediction capacity of the GEP generated models in the context of other commonly utilized MLMs was assessed as well. KRR, ANN, SVM and, RF were used for generating 113 predicted data points as described in section 3.2 (Fig. 12). The prediction performance of GEP, KRR, ANN, SVM and, RF are shown in Fig. 13. Panel a contains the average MEF values computed

for all MLM methods predicted values when compared to the original observations for $R_{eco}, R_{above}, R_{soil}, R_{root}, R_{myc}, R_{soil_a}, R_{soil_h}$. For all other cases, the performance is in the same range for all methods, but the GEP derived models having the lowest mean MEF values. Panel b shows that when all MLM were trained on the residuals obtained from comparing the GEP outputs with the observations, the GEP approach has the lowest capacity of capturing new relevant signals and is strongly outperformed by the rest of the MLM, indicating that amount of information retrievable by GEP with the current fitness and settings is limited

and captured already in the first run.

### 4.2.2   Comparing with literature models

Lastly, the GEP generated models were compared with some of the most commonly used literature models for describing respiration. The resulting MEF values obtained after individual parameter optimization using the CMA-ES procedure for each literature model are given in Tab. 5. The literature model structure that performed best overall in terms of prediction capacity

measured as MEF is the $WaterQ_{10}$ model (Fig. 14). Figure 14 shows as well that certain types of respiration are easier to represent by all models, including the models GEP generated, whilst other types of respiration are poorly predicted by all models. Nevertheless, for all respiration types, the highest MEF values are generally recorded by the GEP models.

    As the studied literature models performed best in modelling $R_{soil}$, we focus on contrasting GEP model results to literature model outcomes for this ecosystem respiration component. Of all models included, the GEP model and $Q_{10}$ model including

$SWC$ dependency captured seasonal variability best, but no model satisfactorily represented short-term $CO_2$ flux variations (Fig. 15, panel a). All models show the largest range of residuals for the months May to July in 2008, and June/July in 2009 (Fig. 15, panel b), with the two best-performing models (GEP and $WaterQ_{10}$) having the narrowest range of absolute residuals. Monthly mean average errors (MAE) indicate as well a systematic underestimation of soil $CO_2$ efflux in the first year (Fig. 4 of suppl.).

## 5   Discussion

### 5.1   On the GEP method

In this work, the primary reason for the artificial experiments was obtaining a better understanding of the capacity of GEP to solve symbolic regression types of problems. We put an emphasis on GEP performance in the presence of noise. This aspect was important, given that monitoring data from terrestrial ecosystem $CO_2$ effluxes are typically contaminated by sometimes

5     substantially large random uncertainties and measurement noise. In the case of NEE flux measurements, Lasslop et al. (2008)

and Richardson et al. (2008) show that the measurement error typically scales with the magnitude of the flux, leading us to simulate that type of situation by adding noise that scales with signal to an already known function, equation 15. The results show that all the studied methods are stable to presence of noise in the training set. These results increase our confidence in the predictions generated by studied machine learning methods; in particular GEP derived modes can tolerate SNRs of 1. Considering that the SNR in the $R_{eco}$ observations (if noise is only considered as random error) is probably larger than 4 which is where the curve starts decreasing in Fig. 4, the noise presence in the data should not influence the automated model construction process and the real signals should be accurately captured when data uncertainties follow the pattern described here. On the other hand, for $R_{soil}$ and other $CO_2$ fluxes measured with other techniques the magnitude and the distribution of the uncertainty can be different (Ryan and Law, 2005; Pérez-Priego et al., 2015), and we cannot state what the response of the present MLM is in the presence of different types of uncertainties and measurement noise.

Our findings illustrate that the selection of CEM over MEF as a fitness function for optimization has a minor effect on the global mean MEF (Fig. 5). We also notice that due to applying constraints on the presence of structure in the residuals and the length of the parameter vector, the final mean number of parameters is lower when CEM is chosen.

### 5.1.1 Limitations

One of the critical aspects in our work is that GEP, as implemented here, can only represent and derive "$n \to 1$" type of response functions. We are not able to generate model structures that encode e.g. system-intrinsic dynamics like feedback loops, which are expected from our current understanding of biogeochemical cycles in terrestrial ecosystems (Ehrenfeld et al., 2005; Friedlingstein et al., 2006). Hence, we believe GEP is suitable to e.g. understand and describe the sensitivities and non-linear responses to changes in hydro-meteorological drivers, but fails to represent more complex carbon or soil water dynamics. Pools and pool transfers cannot be introduced currently in the input, unless the inflow/outflow equations are known and can be included in the set of functions that can participate in the evolution.

Lagged responses can only be detected if the number of lags from a driver is correctly included in the input, which already implies sufficient knowledge of their existence and behaviour. Whilst in the current implementation of the GEP algorithm, shifts in conditions and responses cannot be encoded or detected; these could be addressed with the inclusion of a conditional operator in the set of functions encoded in the GEP evolution individuals.

Nevertheless, it would be fair to mention that the same limitations can affect the results of the other MLM and empirical models presented in this paper. A clear advantage ANN, RF and SVM have though over the GEP symbolic regression construction, is the fact that when the target variable presents a skewed distribution, log-transforming of the target data is recommended for regression type of methods, such as GEP Keene (1995), whereas there is no effect on the prediction capacity of the other MLM as far as we are aware. Moreover, such a log-transformation needs a back-transformations that might induce a bias if the right correction is not performed Manning (1998). For these reasons, in cases where less steps in obtaining predictions are desired and no mathematical expression of the models needed to obtain the predictions are needed, non-GEP approaches might be recommended.

## 5.2 The value of GEP for modelling ecosystem respiration fluxes

We automatically generated a series of model structures to describe terrestrial $CO_2$ respiration fluxes (equations 22-28) with the GEP approach. Most of these structures (5 out of 7) were of rather low complexity, requiring only 4 free parameters and allowing for further interpretation. The most complex structure is found for the $R_{myc}$ representation, which is in line with previous findings (Shi et al., 2012).

Interestingly, the models derived for $R_{eco}$ and $R_{soil}$ are structurally very similar. That is also the case of $R_{root}$ and heterotrophic respiration, where the difference lies in the set of parameters and the added presence of an intercept in the formulation of the $R_{soil_h}$ model. This finding suggests a consistency in the response of the $R_{soil}$ components to their drivers, considering that the separation of the $R_{soil}$ into its components might still lack accuracy (e.g. Hanson et al., 2000; Kuzyakov, 2006; Subke et al., 2006; Heinemeyer et al., 2011).

When we compared the GEP-derived models with the community established semi-empirical models from a structural point of view, we found that they shared some key features for temperature dependencies of $CO_2$ fluxes, which are typically captured by exponential relationships, but reveal some previously unconsidered dynamics as well.

A major difference was in the response of the respiration components to $SWC$, where the GEP models often chose $SWC$ as one of the drivers. Moreover, the GEP models often contained an exponential dependency, i.e. there are only certain parts of the signal that are strongly sensitive to varying $SWC$. We believe that the exponential dependency of terrestrial ecosystem respiration components to $SWC$ is a very intuitive pattern that has not yet been reported in the literature, and requires further exploration.

Another difference we found was the strongly seasonal response of the respiration components to $GPP$, possibly as a proxy to light and vegetation availability which were not included in the set of candidate predictors.

Considering that GEP identified plausible models, that are very different structurally from previously reported semi-empirical models, still yielding equivalent or better modelling performance, the validity of the conventional semi-empirical models can be questioned. Nevertheless, we do believe that there is need for more in-depth analysis for determining whether the GEP described processes make actual biological sense and the selected drivers and their interactions represent true processes and responses.

## 5.3 Data quality

During our study, it was apparent that the highest MEF values were obtained for all the studied methods in the case of the respiration types that had direct measured observations and were not derived. It might be the case that when fluxes are obtained from derivations, the measurement error will also increase, and the partition of clear signal existing in the observations is not sufficient for constructing a good model with GEP.

## 5.4 High frequency variability

All GEP generated models underestimated the high respiration fluxes (Fig. 8) and typically did not capture the fast responses.This phenomenon was in some cases a systematic pattern, and sometimes affected only certain times of the year. Similarly, semi-empirical models struggled to adequately simulate $CO_2$ flux peaks and in some cases monthly flux averages (Fig. 15).

A more in-depth comparison of all the GEP and conventional respiration models, based on a time-scale dependent assessment of model-data mismatch (Mahecha et al., 2010) could help to further elucidate the problem and clarify some of the strengths and weaknesses of the different modelling approaches, especially when seasonal mismatches appear. Nevertheless, a detailed time-scale dependent assessment is beyond the scope of this study, and for such an analysis, the current time series are simply too short.

The question is whether the GEP method lacks the ability to build models that correctly represent the processes and their fast dynamic responses, or whether the candidate drivers and the observations used for their representation are simply not sufficient for generating representative models. In the end, the response of $R_{soil}$ and $R_{eco}$ to external drivers might be too complex to describe solely with the currently available measurements and with the selected drivers.

We believe that the consistent underestimation of fast responses was partly due to surface moisture affecting litter decomposition and fungal activity, as soil moisture was only monitored over the average 8 cm surface, with the top few centimetres most likely presenting the highest activity and partly due to some potential processes/drivers like lags between $GPP$ and respiration (Hölttä et al., 2011) or phenology (Migliavacca et al., 2015) that were not specifically included in the learning process.

Another explanation for missing some of the (high flux) variability could be in our choice of fitness function. As we decided to penalize during the learning process for structures with many parameters, it is likely that some structures were eliminated early-on during this process, even though they may be well-suited for describing a given process from a modelling efficiency point of view. However, this is a case of trade-off between a good fit and structural simplicity, and in our approach, we decided that simplicity of structure, i.e. the possibility of interpretation is a very important asset.

We explored as well the possibility of the underestimation of the carbon flux variability being caused by the log-transformations applied to the observations. It could have been the case that the log-transformations excluded interesting components of the model structures by forcing the method to build multiplicative models. Nevertheless, when the GEP was run again on the residuals, without log-transforming, no new meaningful information was retrieved, indicating that multiplicative models were sufficient for reconstructing the $R_{eco}$ components present in this study.

## 5.5 Equifinality

Table 4 shows that when optimizing the parameters for all structures, the prediction performance becomes similar, which leads to the question of equifinality of dynamical systems, where different models that try to capture their structure, might have different formulations, but represent the same response.

A critical question for the applicability of any ecosystem model is whether the model structure is more important than the parametrisation of a given "best" model. For this question to be addressed, however, we need a larger sample of ecosystem types representative for different types of responses where we can explore the importance of the obtained structures and their parameter sets.

## 5.6 GEP models in the context of other machine learning methods

The comparison of GEP generated models and machine-learning methods showed a narrow range of predicted fluxes (Fig. 13). The analysis of training all the MLM on the GEP residual output showed that the GEP approach is not able to retrieve any new meaningful structural components, but that the remaining MLM are much better at reconstructing the signal left in the residuals. This indicates that although the GEP is actually a reliable MLM when it comes reconstructing the underlying $R_{eco}$ fluxes and is not prone to over-fitting, it could be that the current set-up of the GEP is not sufficient for an exhaustive description of those fluxes, or that might be overly strict on complexity of models compared to other MLM. The GEP approach has, nevertheless, the benefit of producing mathematical model structures that can be the basis for future interpretation.

## 6 Conclusions and Outlook

Overall, our results suggest that the GEP approach is a potentially powerful tool of reverse engineering, particularly helpful for building ecological models when there is a minimum of a priori system understanding. We exemplified this conceptually using artificial data, but also show that GEP always yields as good or better results compared to conventionally used models in the case of ecosystem respiration. Based on data from a long-term monitoring site of different respiratory fluxes, and using GEP as a reverse engineering tool, we found new structures for modelling $R_{eco}$ components. The GEP derived models outperform conventionally used models and generally differ by the way temperature and $GPP$, but also $SWC$ are interpreted, indicating that conventional respiration models might have to be revised. At the same time, we found that when the GEP derived models are mutually compared, there are sufficient structural particularities for each terrestrial respiration type as to not allow for the formulation of a general $R_{eco}$ law. More research is needed on a larger set of sites to identify widely usable models and for their interpretation. A particular matter of concern is the apparent equifinality of selected model structures, indicating that many response functions are yielding predictions of almost similar quality. A study of multiple sites would enable an investigation of whether specific ecosystem types result in similar model structures, or whether response functions apply across contrasting ecosystem types.

The current study has also revealed methodological aspects that could be improved. In particular, we found the inclusion of a parameter optimization step very helpful to further test the transferability of model structures. But this approach could be potentially integrated into the GEP evolution. More specifically, we think that the next development of GEP could include the parameter optimization as an intermediate step before selection during each evolution generation (Ilie et al.). In this way, a model structure could be chosen according to not only the current state of parameters but also on its potential and convergence to a global solution might be achieved faster.

## Code and data availability

All code and data used to produce the results of this paper can be provided upon request by contacting Iulia Ilie or Miguel D. Mahecha.

## Acknowledgements

We thank Markus Reichstein for all the useful comments and suggestions.

This work was supported by the International Max-Planck Research School for global Biogeochemical Cycles (IMPRS-gBGC), Jena, by the European Union's H2020 research and innovation programme project BACI; grant agreement 640176 and by NOVA grant UID/AMB/04085/2013. The Alice Holt Forest GHG Flux site is funded by the UK Forestry Commission.

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

## Glossary

20    **chromosome**   individual used in automatically evolving an optimal solution comprised of a set of genes that are connected with a binary operation (e.g. $+ \times -$). 5

     **CMA-ES**   covariance matrix adaptation evolutionary strategy. 8

     **evolution**   the process of producing an optimal solution by GEP through . 4

     **expression tree**   binary tree used to represent algebraic expressions. 5

25    **gene**   set of characters of fixed length that encodes an expression tree. 5

     **gene head**   initial section of the string that comprises a GEP gene, containing a combination of characters that map to predictors and possible functional transformations . 5

     **gene tail**   end section of the string that comprises a GEP gene, containing only characters that map to predictors. 5

     **generation**   time step of an evolution. 5, 6

     **genetic operator**   operator that produces changes in the structure of a chromosome and the expression tree it encodes by

5       altering the strings representing composing genes (e.g. mutation, inversion, recombination, etc.) . 6

     **genetic operator rate**   probability of a genetic manipulation to occur during a generation. 6

     **GEP**   **g**ene **e**xpression **p**rogramming, machine learning method that evolves chromosome structures with the purpose of minimizing a cost function. 3

     **hyper-parameter**   set of parameters which need to be set for the runs of a machine learning approach. 6

10   **ill-posed problem**   a problem for which the solutions might not be unique or unstable, also known as an inverse problem. 8

     **individual**   GEP entity that is a component of a population during a certain step of the evolution process. Also known as chromosome. 6

     **MLM**   **m**achine **l**earning **m**ethod that can produce predicted values based on a training set. 10

     **population**   total set of chromosomes that participate at a certain step in the evolution of an optimal solution in the GEP approach.. 5

     **reproduction**   process of generating new individuals for a new generation starting from the present generation individuals after they go through structure modification and fitness based selection. 5

840   **solution**   finally selected model structure resulting from a GEP run. 3

**Table 1.** GEP settings

| Parameter | Artificial data | Real observations |
|---|---|---|
| Number of chromosomes | 2000 | 2000 |
| Number of genes | 3 | 2 |
| Head length | 5 | 6 |
| Functions | $+,-,/,*,x^y,\sqrt{},\ln,\exp,\sin,\cos$ | $+,-,/,*,x^y,\sqrt{},\ln,\exp$ |
| Terminals | $x_1,x_2,x_3$ | $GPP_s, T_{Air}, T_{-10}, SWC$ |
| Link function | + | + |
| Max run time | 1200 seconds | 1800 seconds |
| Fitness function | CEM | CEM |
| Selection method for replication | tournament(Coello and Montes, 2002) | tournament |
| Mutation probability | 0.2 | 0.2 |
| IS and RIS transpositions probabilities | 0.05 | 0.05 |
| Two-point recombination probability | 0.3 | 0.3 |
| Inversion probability | 0.05 | 0.05 |
| One point recombination probability | 0.4 | 0.4 |

**Table 2.** Respiration model formulations commonly used in the environmental science community

| Model | Formulation | Reference |
|---|---|---|
| Arrhenius | $a \times e^{-E_0/RT}$ | (Lloyd and Taylor, 1994) |
| $Q_{10}$ | $\phi_1 \times \phi_2^{(\frac{T-T_{ref}}{10})}$ | (Reichstein and Beer, 2008) |
| Water $Q_{10}$ | $\phi_1 \times \phi_2^{(\frac{T-T_{ref}}{10})} \times \frac{SWC}{SWC+\phi_3} \times \frac{\phi_4}{SWC+\phi_4}$ | (Richardson et al., 2008) |
| $LinGPP$ | $(R_0+k_2GPP) \times e^{E_0(\frac{1}{T_{ref}-T_0}-\frac{1}{T_A-T_0})} \times \frac{\alpha k+SWC(1-\alpha)}{k+SWC(1-\alpha)}$ | (Migliavacca et al., 2011) |
| $ExpGPP$ | $[R_0+R_2(1-e^{k_2GPP})] \times e^{E_0(\frac{1}{T_{ref}-T_0}-\frac{1}{T_A-T_0})} \times \frac{\alpha k+SWC(1-\alpha)}{k+SWC(1-\alpha)}$ | (Migliavacca et al., 2011) |
| $addLinGPP$ | $R_0 \times e^{E_0(\frac{1}{T_{ref}-T_0}-\frac{1}{T_A-T_0})} \times \frac{\alpha k+SWC(1-\alpha)}{k+SWC(1-\alpha)} + k_2GPP$ | (Migliavacca et al., 2011) |
| $addExpGPP$ | $R_0 \times e^{E_0(\frac{1}{T_{ref}-T_0}-\frac{1}{T_A-T_0})} \times \frac{\alpha k+SWC(1-\alpha)}{k+SWC(1-\alpha)} + R_2(1-e^{k_2GPP})$ | (Migliavacca et al., 2011) |

$a, E_0, \phi_1, \phi_2, \phi_3, \phi_4, R_0, R_2, k, k_2$ and $\alpha$ are model parameters that can be optimized

**Table 3.** Modelling performance for all extracted model structures after cross validation over 90 cases.

| Respiration type | $MEF$ | $\sigma$MEF | Equation |
|---|---|---|---|
| $R_{eco}$ | 0.57 | 0.13 | 22 |
| $R_{above}$ | 0.31 | 0.23 | 23 |
| $R_{soil}$ | 0.79 | 0.04 | 24 |
| $R_{root}$ | 0.59 | 0.08 | 25 |
| $R_{myc}$ | 0.39 | 0.28 | 26 |
| $R_{soil_a}$ | 0.82 | 0.05 | 27 |
| $R_{soil_h}$ | 0.52 | 0.08 | 28 |

**Table 4.** Average validation MEF performance for all extracted model structures when re-optimized against all other respiration $CO_2$ flux observations.

| trained for/ opt. for | $R_{eco}$ | $R_{above}$ | $R_{soil}$ | $R_{root}$ | $R_{myc}$ | $R_{soil_a}$ | $R_{soil_h}$ |
|---|---|---|---|---|---|---|---|
| $R_{eco}$ (Eq. 22) | 0.57 | 0.27 | 0.77 | 0.58 | 0.10 | 0.68 | 0.42 |
| $R_{above}$ (Eq. 23) | 0.56 | 0.31 | 0.69 | 0.44 | 0.07 | 0.60 | 0.46 |
| $R_{soil}$ (Eq. 24) | 0.50 | 0.20 | 0.79 | 0.47 | 0.38 | 0.82 | 0.39 |
| $R_{root}$ (Eq. 25) | 0.23 | 0.27 | 0.57 | 0.59 | 0.01 | 0.65 | 0.51 |
| $R_{myc}$ (Eq. 26) | 0.54 | 0.22 | 0.82 | 0.50 | 0.39 | 0.84 | 0.52 |
| $R_{soil_a}$ (Eq. 27) | 0.50 | 0.20 | 0.79 | 0.47 | 0.38 | 0.82 | 0.39 |
| $R_{soil_h}$ (Eq. 28) | 0.55 | 0.26 | 0.76 | 0.56 | 0.06 | 0.67 | 0.52 |

**Table 5.** Average validation MEF performance for CMA-ES optimized selected literature model formulations when compared with respiration $CO_2$ flux observations.

| Model formulation | $R_{eco}$ | $R_{above}$ | $R_{soil}$ | $R_{root}$ | $R_{myc}$ | $R_{soil_a}$ | $R_{soil_h}$ |
|---|---|---|---|---|---|---|---|
| Arrhenius | 0.41 | 0.15 | 0.65 | 0.50 | 0.07 | 0.61 | 0.38 |
| $Q_{10}$ | 0.47 | 0.19 | 0.69 | 0.52 | 0.09 | 0.62 | 0.46 |
| Water $Q_{10}$ | 0.50 | 0.20 | 0.79 | 0.55 | 0.40 | 0.81 | 0.43 |
| $LinGPP$ | 0.55 | 0.25 | 0.74 | 0.57 | 0.17 | 0.70 | 0.49 |
| $ExpGPP$ | 0.58 | 0.30 | 0.76 | 0.57 | 0.20 | 0.72 | 0.54 |
| $addLinGPP$ | 0.55 | 0.27 | 0.73 | 0.56 | 0.12 | 0.67 | 0.48 |
| $addExpGPP$ | 0.56 | 0.27 | 0.73 | 0.54 | 0.20 | 0.69 | 0.49 |

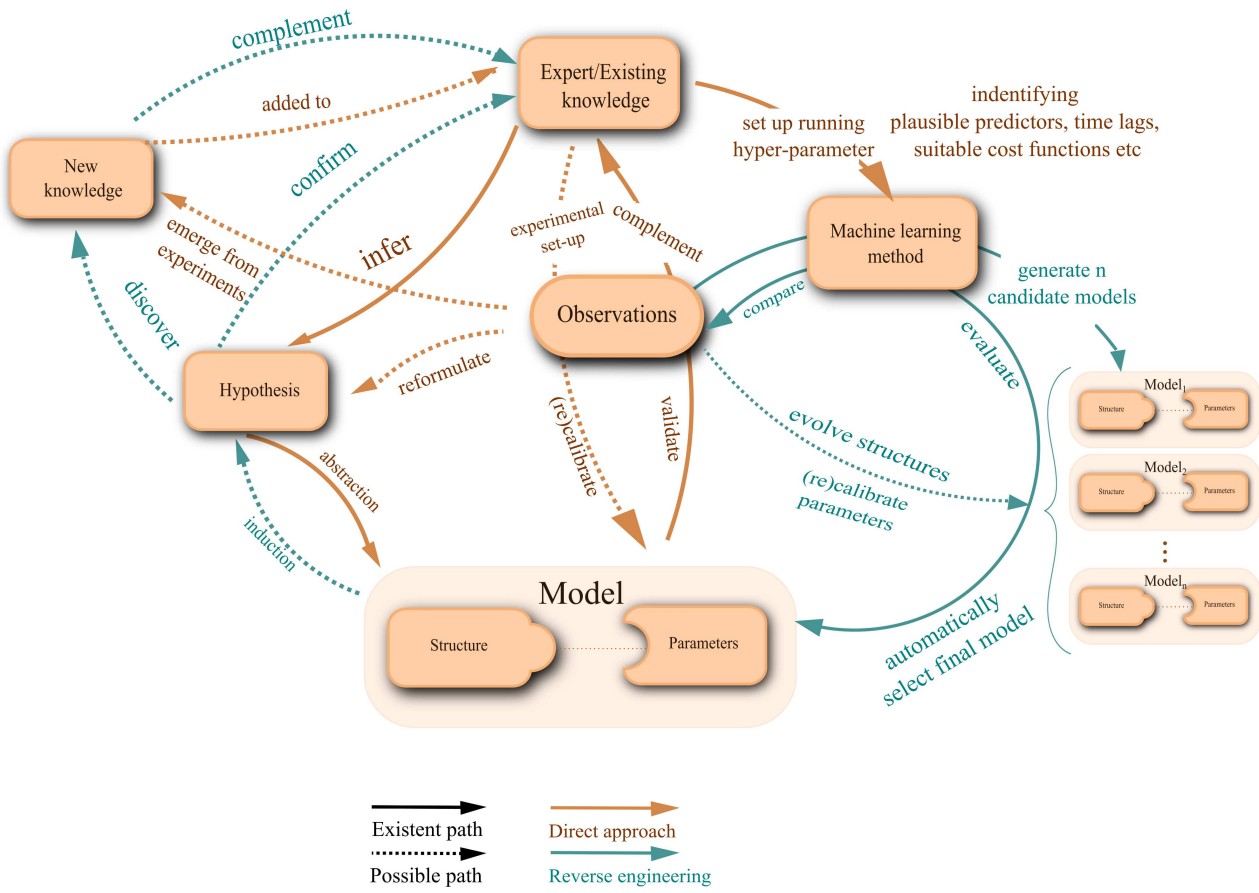

**Figure 1. Direct approach and reverse engineering in model development for describing dynamical systems.** Existing and possible steps needed in the process of building a model. For the direct approach, the process starts with the building of hypothesis from existing knowledge, the hypothesis is then subject of abstraction and summarized in a mathematical model that has two components: the structure and the parameters. The mathematical model can be translated into a computational form that will generate predictions. Depending on how well the predicted values manage to recreate the available observations, the model's parameters are calibrated or if the general trends are missed, there might be need for structural reformulation. On the other hand, in the reverse engineering approach, a machine learning method is used to generate a set of candidate models that are then compared with the available observations and which according to the prediction capacity may have to go through structural changes by automatic evolution or through a final parameter adaptation. From the set of evolved models, the best model in terms of prediction capacity is chosen and its structure will be the basis for hypothesis building, as an expert would try to explain why a specific structure was automatically evolved and whether the structure of the model can be explained from the studied system intrinsic processes. If that will be the case, and the structure has not emerged randomly, the conclusions can be compared with the existing knowledge which can be reconfirmed or new aspects of the studied system might be brought into light.

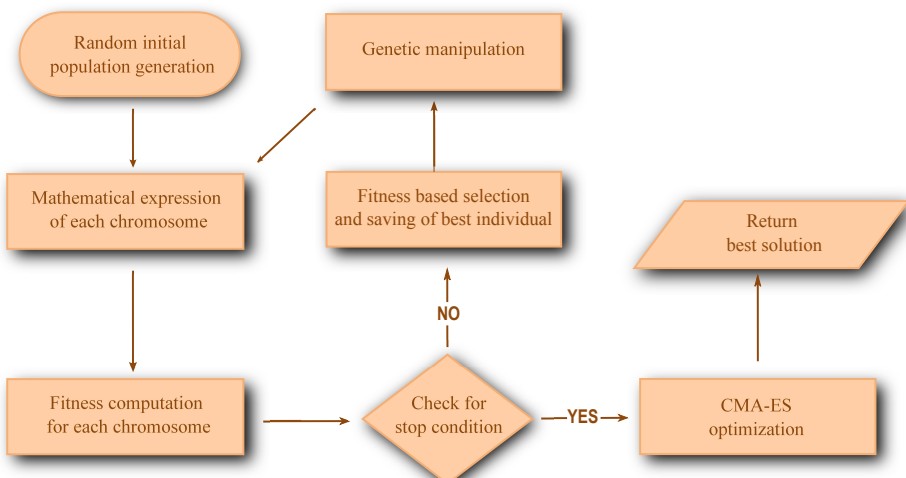

**Figure 2. The work flow used in solving symbolic regression problems with GEP.**. The process of evolving an optimal solution from observations starts with randomly generating a set number of evolution individuals called chromosomes. The chromosomes are composed of genes that are sets of strings encoding expression trees that can be translated into mathematical expressions in the subsequent step. Following the mathematical expression comes the evaluation of each emerging individual (model) against the target variable values and for each one a fitness values is assigned. If the stopping criterion has not been reached (e.g.. best fitness possible, highest number of generations allowed, convergence etc.) the best individual in terms of fitness is saved and the remaining set of chromosomes are selected for genetic manipulation. When the stop criterion is reached, the parameters of the best chromosome is calibrated against the training data with an optimization approach, the CMA-ES, and the best solution is returned.

## Generation

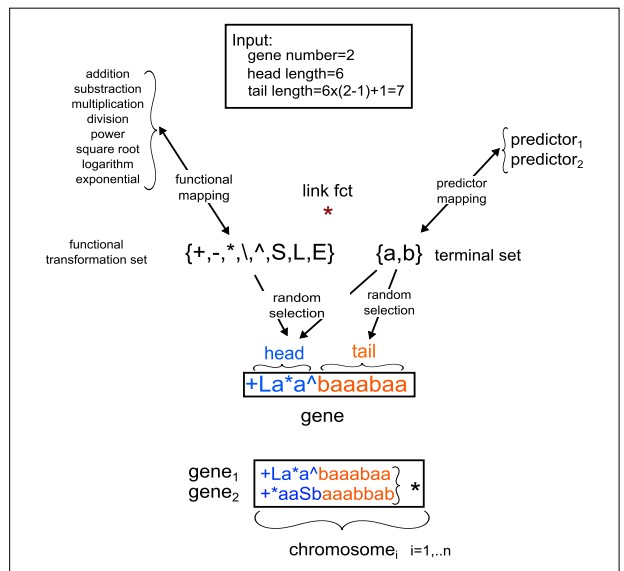

## Translation

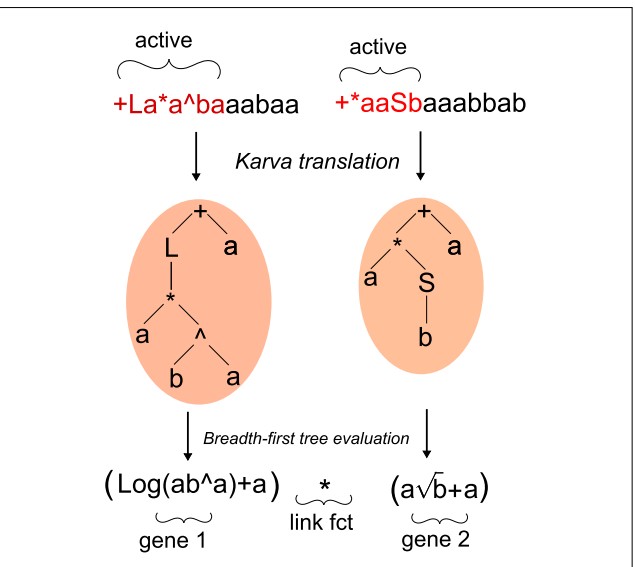

## Mutation

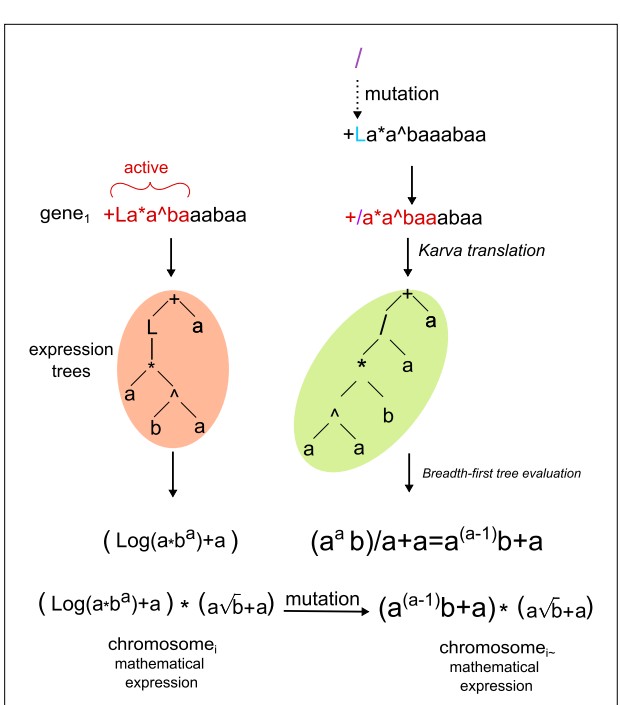

## Genetic operators

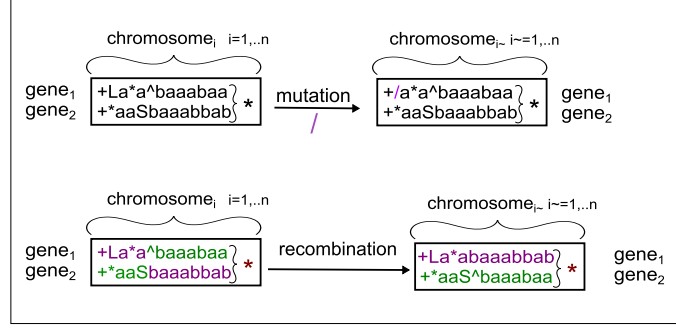

**Figure 3. GEP evolution process components.A.** Initial random generation of genes for creating chromosomes, the individuals evolved by GEP. **B.**GEP internal translation process from strings to expression trees and mathematical expressions.**C.** Changes made in the mathematical expression when applying the mutation operator on the genes of a GEP individual.**D.** Types of genetic operators for changing the GEP evolution individuals.

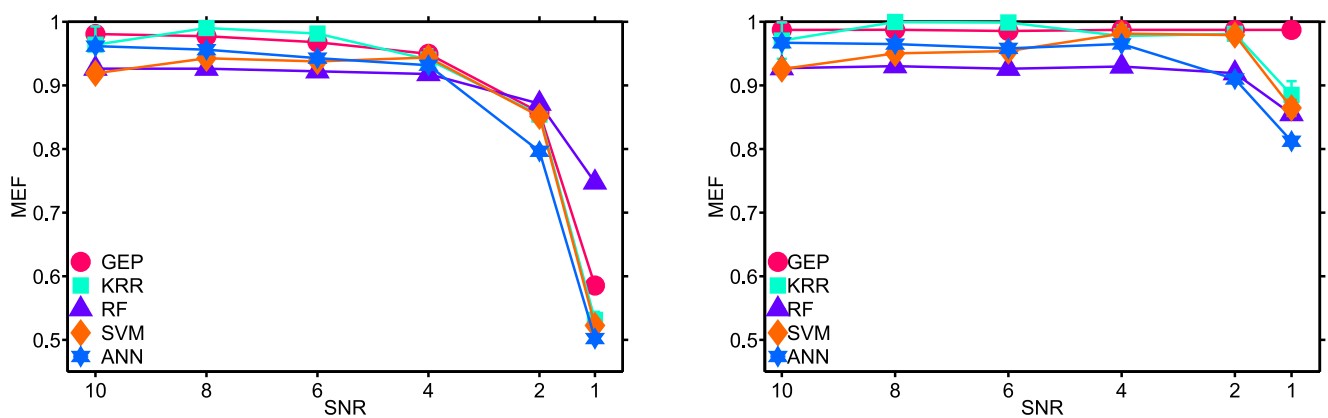

**Figure 4. Effect of adding noise to original signal on prediction capacity for GEP, KRR, RF, SVM and ANN**. The first panel contains the evolution of mean modelling efficiency (MEF) values from 20 independent runs for each increasing level of noise. MEF is computed after learning from a data set of 200 data points and validating against 1000 data points containing noise. The second panel shows the evolution of mean MEF values from 20 independent runs for each increasing level of noise where MEF is computed after learning from a data set of 200 data points and validating against noise-free 1000 data points generated from equation 15.

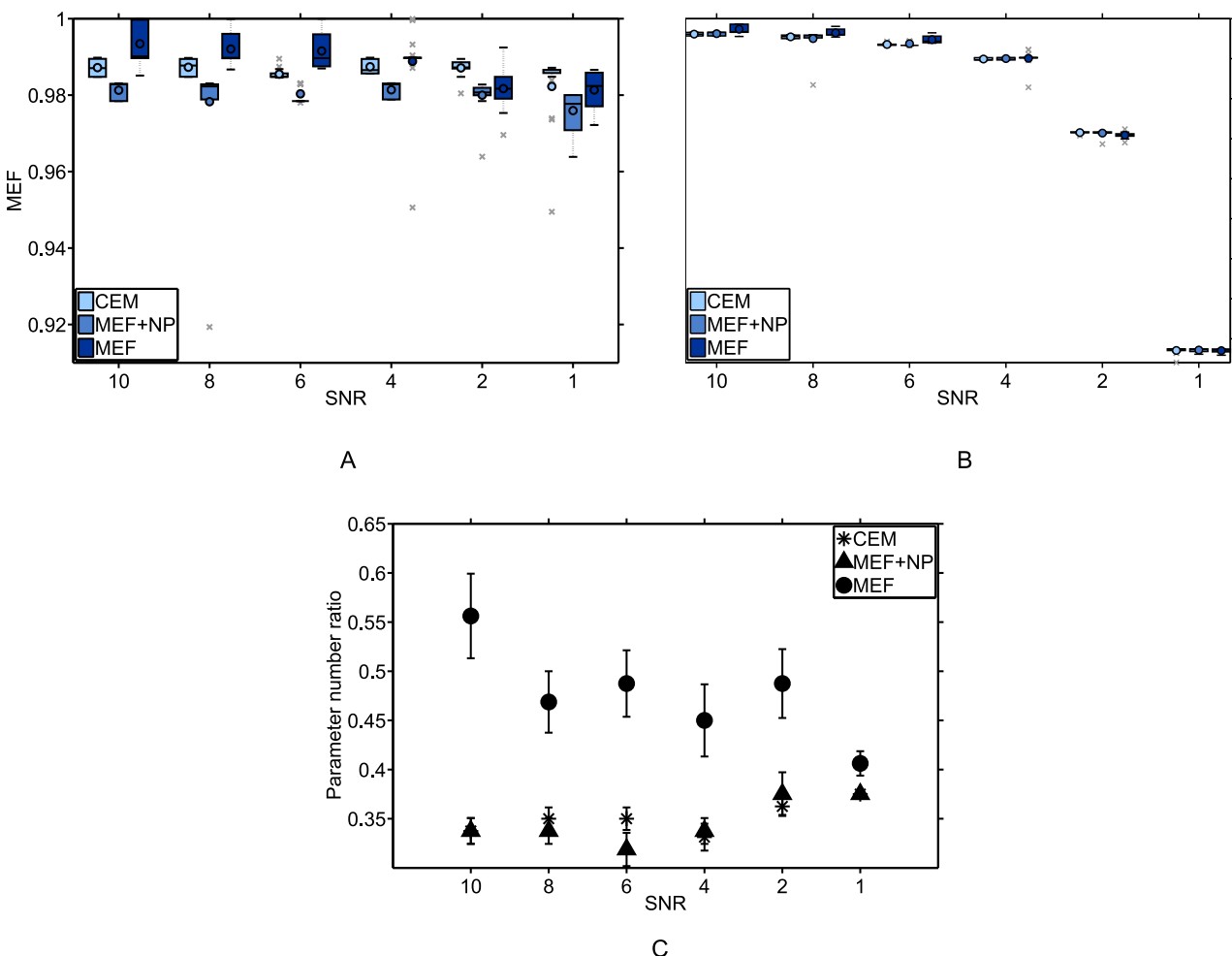

**Figure 5. Effects on modelling performance and parameter number caused by choice of fitness function** during GEP training for artificial noisy data generated by equation 15, where MEF is defined in equation 1 and CEM is defined in equation 3. **A**. Mean MEF when validation against noisy data after 20 GEP runs with different fitness functions. **B**. Mean MEF when validation against noise-free data after 20 GEP runs with different fitness functions. **C**. Ratio of predicted number of parameters to true number of parameters after 20 GEP runs with different fitness functions.

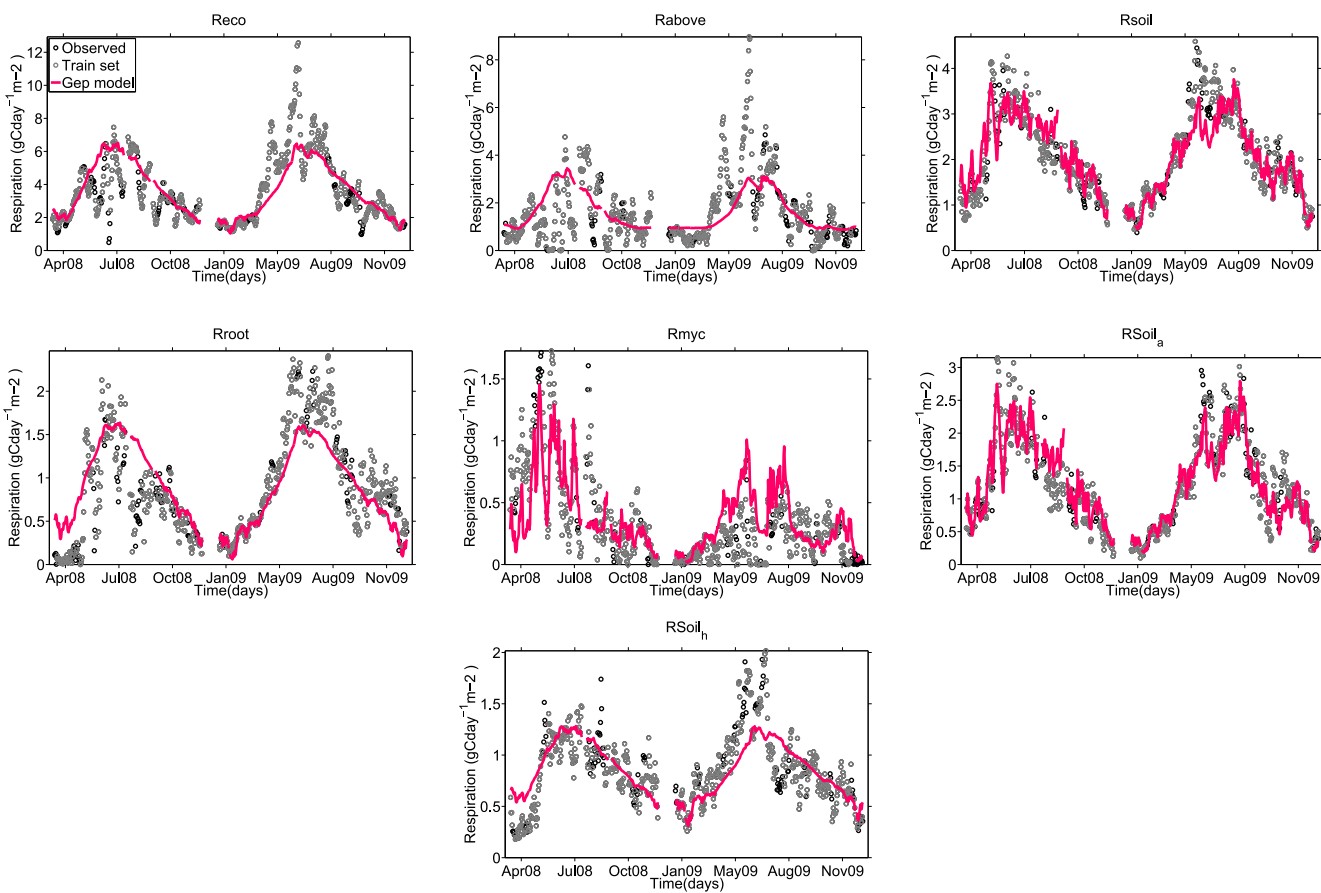

**Figure 6. Observed and predicted outgoing** $CO_2$ **fluxes.** 613 time steps of daily averaged $CO_2$ effluxes for two years at the Alice Holt oak forest site. The predicted values are generated with the models extracted by the GEP approach with the settings given in table 1 for the following types of respiration: $R_{eco}, R_{above}, R_{soil}, R_{root}, R_{myc}, R_{soil_a}, R_{soil_h}$ and back-transformed with a smear term bias correction. The models are given in equations: 22-28

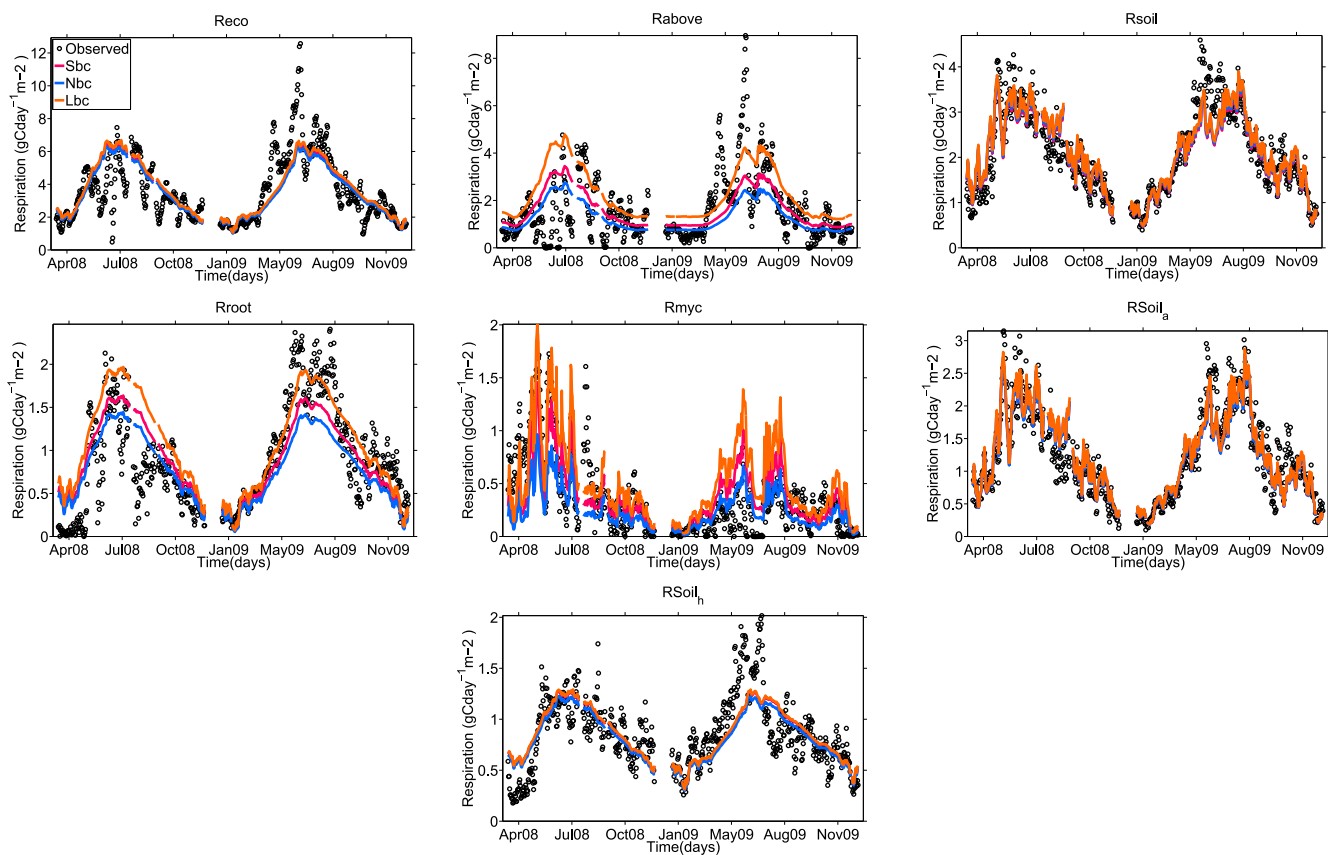

**Figure 7. Observed and predicted outgoing** $CO_2$ **fluxes.** 613 time steps of daily averaged $CO_2$ effluxes for two years at the Alice Holt oak forest site. The predicted values are generated with the models extracted by the GEP approach with the settings given in table 1 for the following types of respiration: $R_{eco}, R_{above}, R_{soil}, R_{root}, R_{myc}, R_{soil_a}, R_{soil_h}$ and back-transformed with 3 types of residual **b**ias **c**orrection terms: **s**mear term , **n**aive, and **l**og normal term.

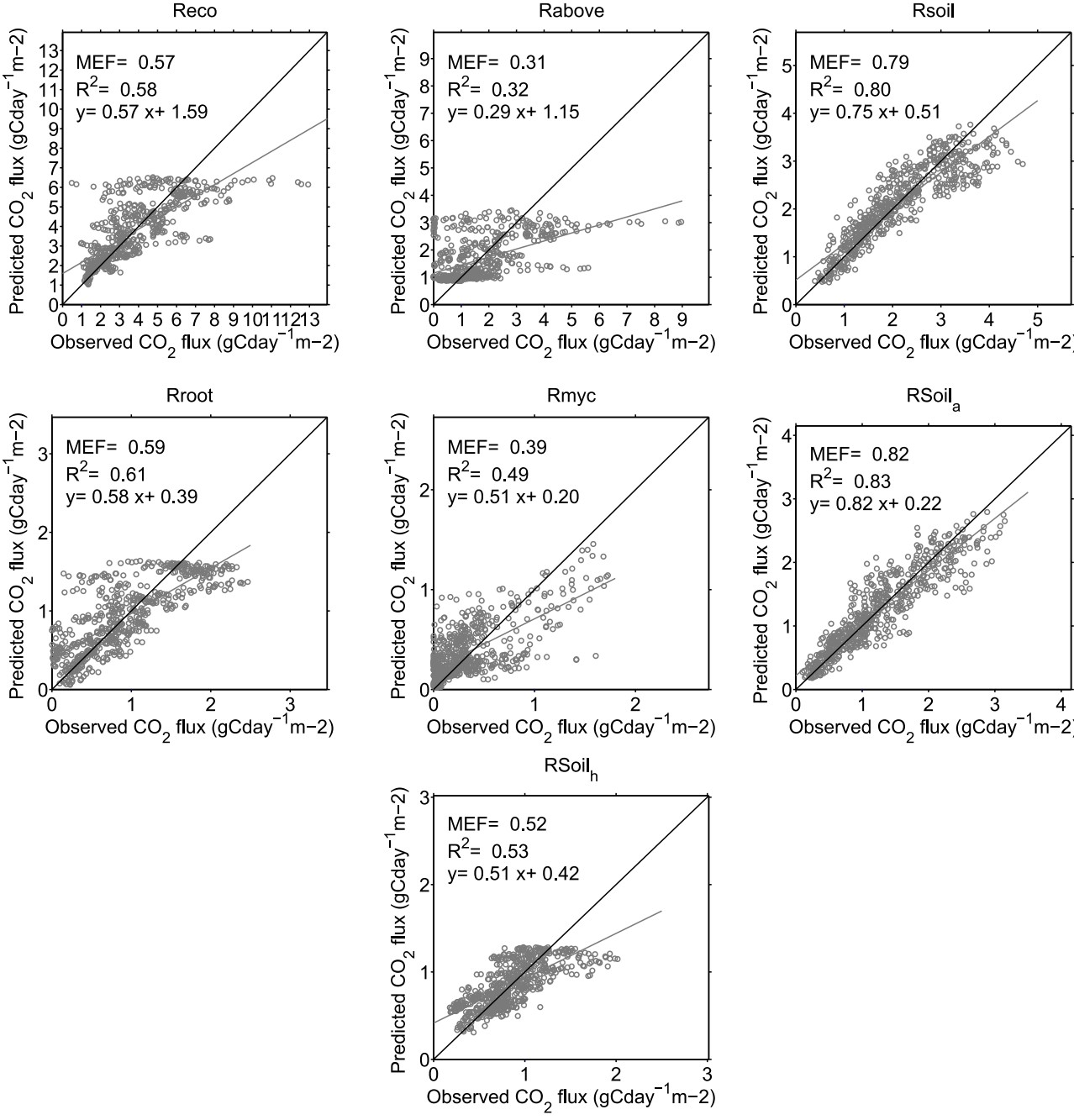

**Figure 8. Observed and predicted outgoing** $CO_2$ **fluxes.** 613 time steps of daily averaged $CO_2$ effluxes for two years at the Alice Holt oak forest site. The predicted values are generated with the models extracted by the GEP approach with the settings given in table 1 for the following types of respiration: $R_{eco}, R_{above}, R_{soil}, R_{root}, R_{myc}, R_{soil_a}, R_{soil_h}$ and back-transformed with a smear term bias correction. The models are given in equations: 22-28

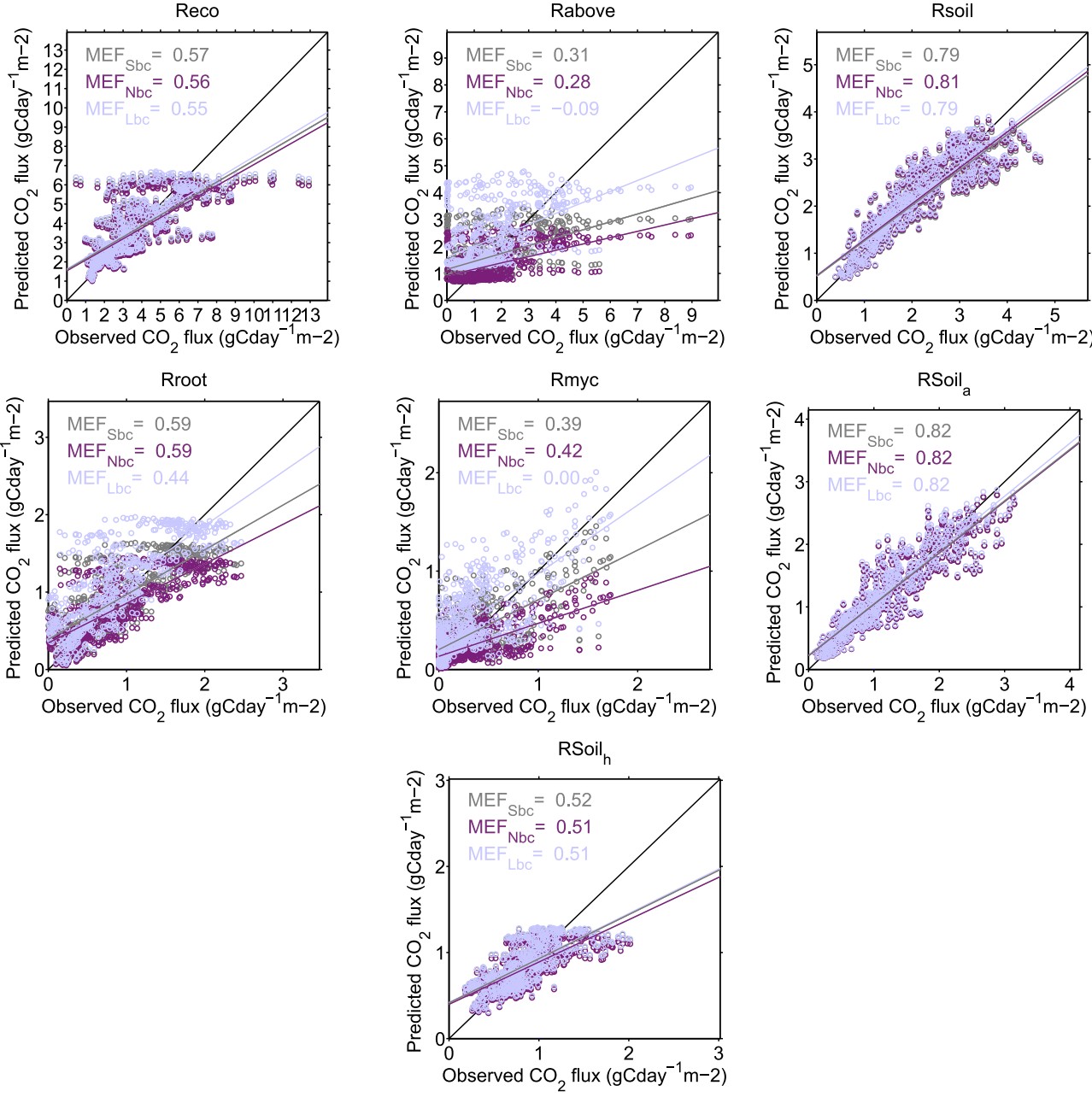

**Figure 9. Observed and predicted outgoing** $CO_2$ **fluxes.** 613 time steps of daily averaged $CO_2$ effluxes for two years at the Alice Holt oak forest site. The predicted values are generated with the models extracted by the GEP approach with the settings given in table 1 for the following types of respiration: $R_{eco}, R_{above}, R_{soil}, R_{root}, R_{myc}, R_{soil_a}, R_{soil_h}$ and back-transformed with 3 types of residual **b**ias **c**orrection terms: **s**mear term , **n**aive, and **l**og normal term. The figure contains the MEF values for each type of bias correction in each respective colour.

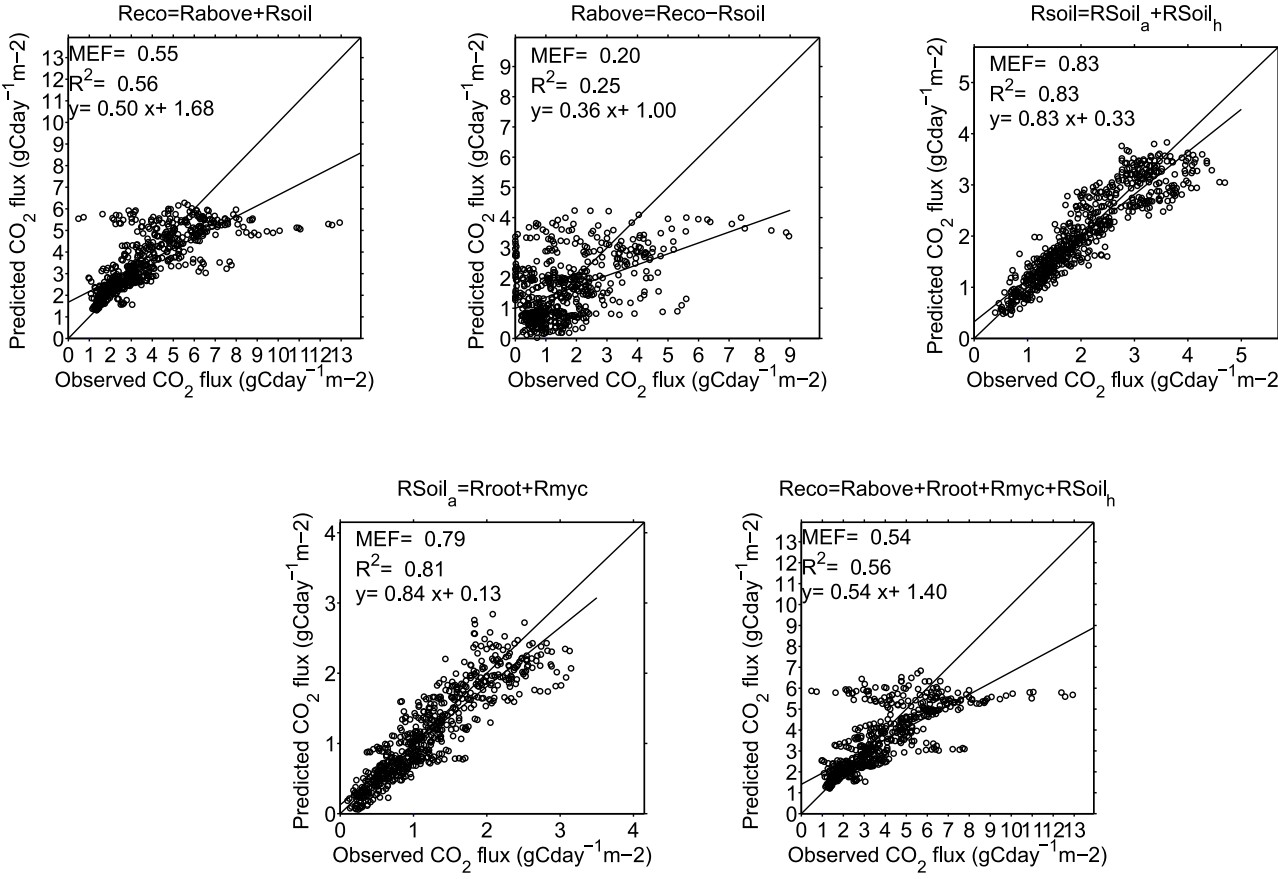

**Figure 10.** Observed versus predicted $R_{eco}$ components fluxes, where predicted values are computed as derived fluxes based on the GEP models given in Eq. 22-28 that were trained on 500 d.p of daily mean values of various $R_{eco}$ components.

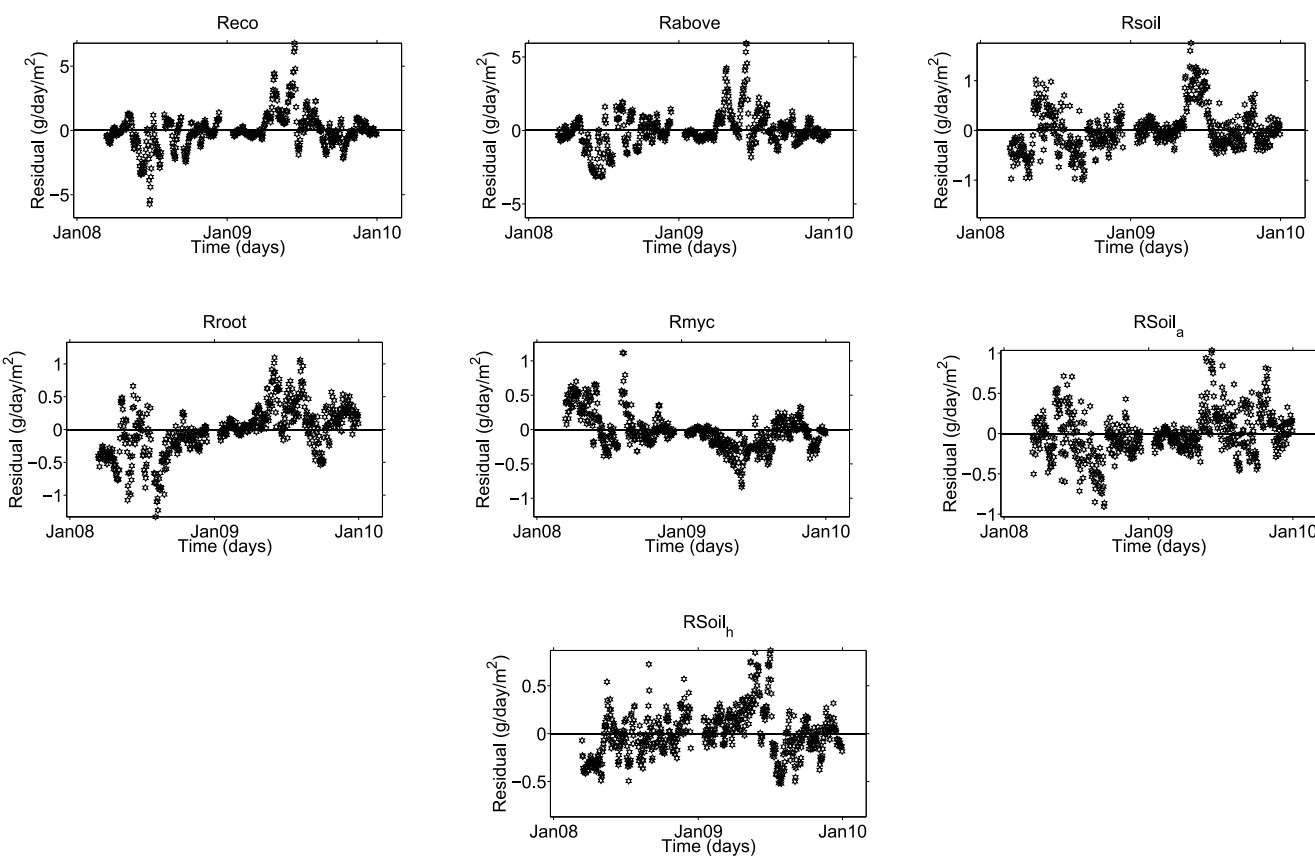

**Figure 11. Residuals computed for smear term bias corrected back-transformed GEP models for various types of** $CO_2$ **respiration fluxes** after training against log-transformed targets with the settings given in column 2 of Tab. 1.

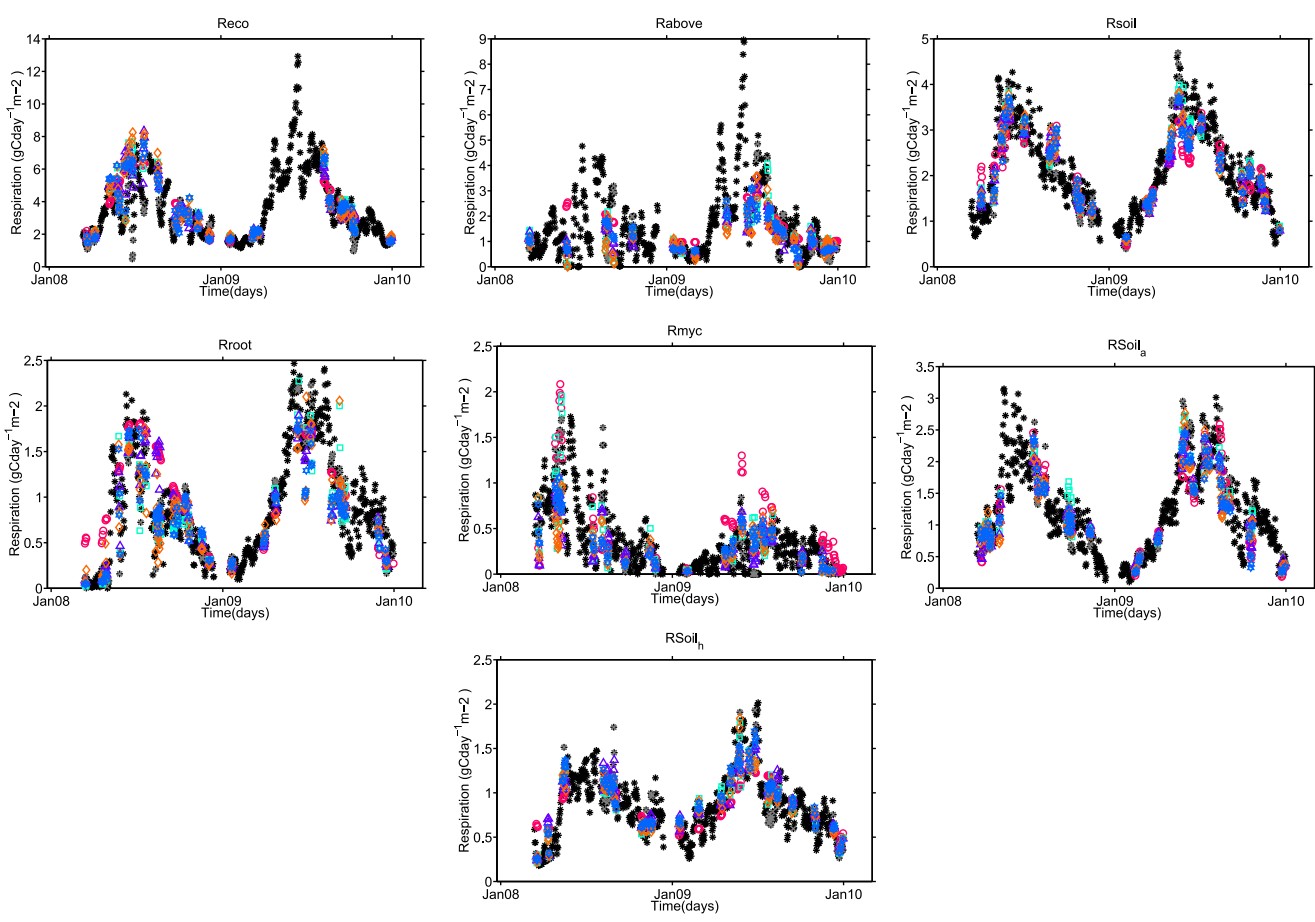

**Figure 12. Observed $CO_2$ fluxes and one set of 113 predicted values given by the some common machine learning methods (MLM) after training on 500 data points and after smear term bias corrected back-transformation.**

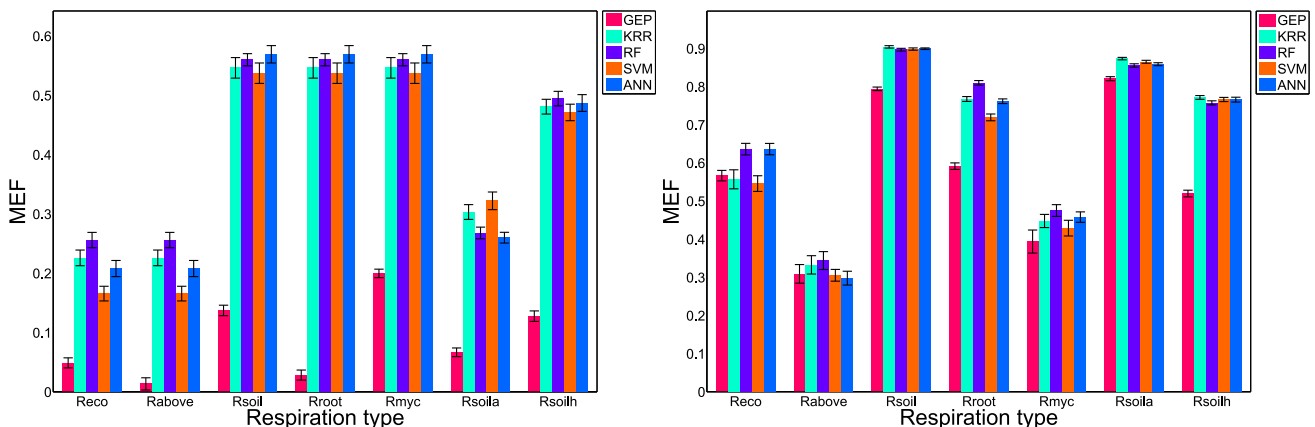

**Figure 13. Machine learning methods (MLM) prediction performance for all respirations components (left) and for the residuals (right) resulting from the GEP trained models after smear term bias corrected back-transformation**. The MEF values obtained for validation by all the MLM methods for $R_{eco}, R_{above}, R_{soil}, R_{root}, R_{myc}, R_{soil_a}, R_{soil_h}$

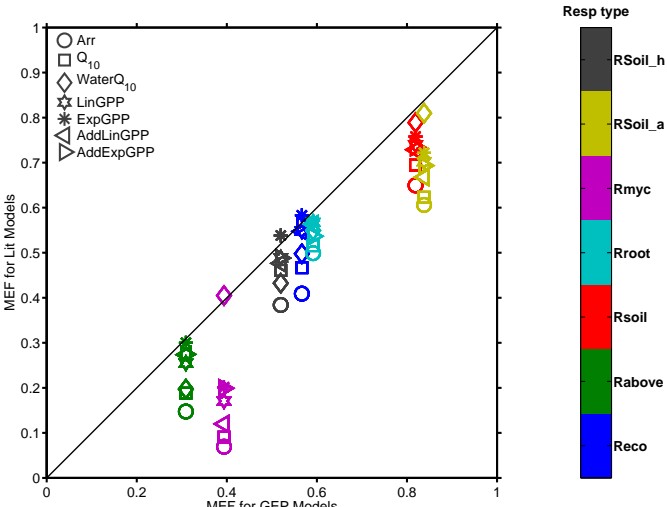

**Figure 14. MEF validation values for literature models and for the best GEP model in terms of MEF at each respiration level.** Each $R_{eco}$ flux component is shown in a separate colour.

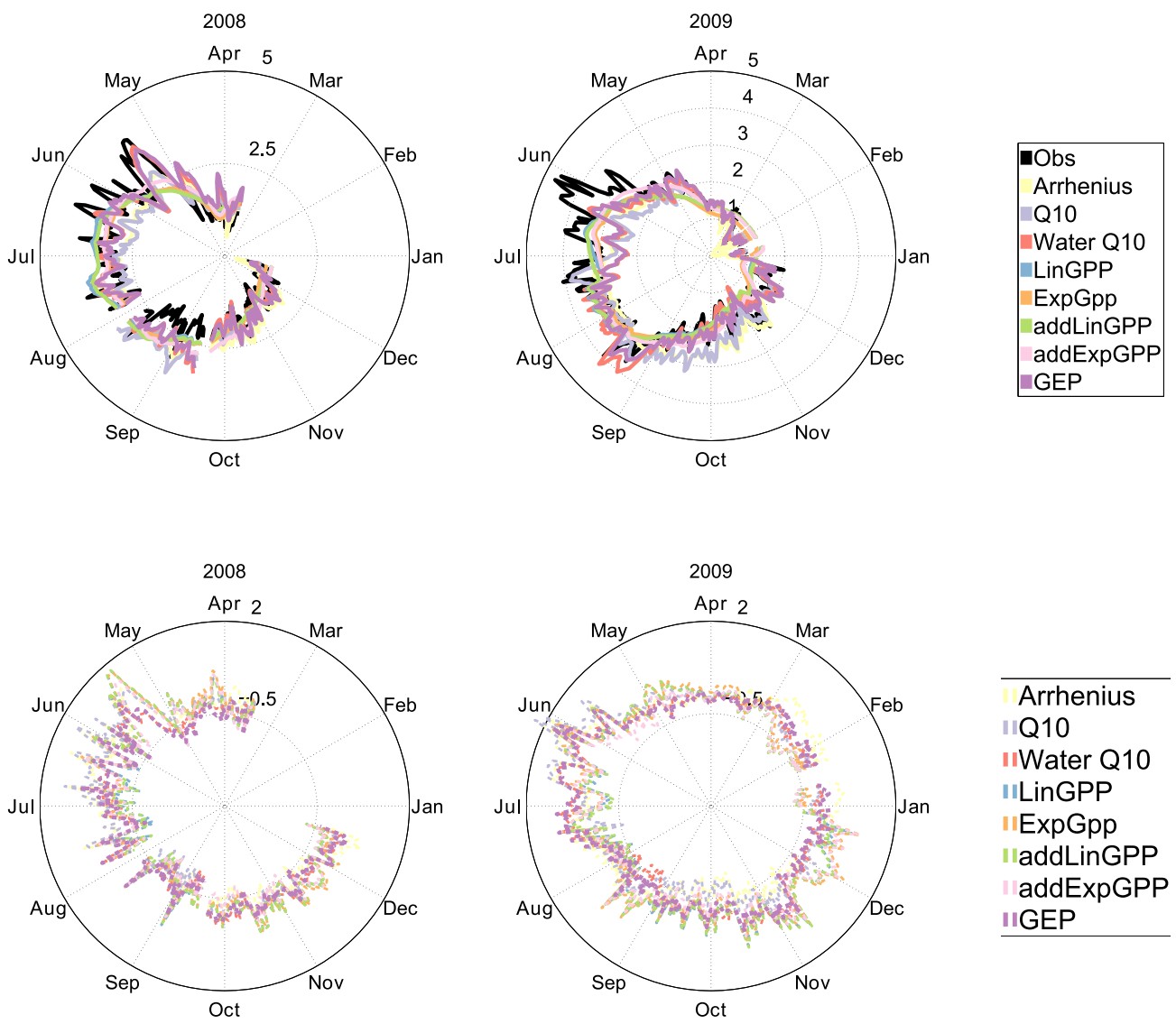

**Figure 15. Daily $R_{soil}$ fluxes (A) illustrated in the context of the two studied years and residual values (B) of the total soil daily $CO_2$ outgoing fluxes as simulated by the investigated literature models and the GEP emerged model after smear term bias corrected back-transformation.** The fluxes shown here are the real flux measured at the site and the predicted fluxes generated according to the GEP model and some of the models used in the environmental science community. The centre of the plots in the second row is -1. The scale of the fluxes is given in gC/$m^2$/day.

## Supplemental Materials: Reverse engineering model structures for soil and ecosystem respiration: the potential of gene expression programming

**Table 1.** Standard error of the MEF at validation values for all MLM for different SNR values when the MEF values are computed against the noisy data.

| SNR | GEP | KRR | RF | SVM | ANN |
|------|------|------|------|------|------|
| 9.82 | 0.00 | 0.00 | 0.02 | 0.00 | 0.00 |
| 8.18 | 0.00 | 0.00 | 0.02 | 0.02 | 0.00 |
| 7.01 | 0.00 | 0.00 | 0.02 | 0.01 | 0.00 |
| 6.14 | 0.00 | 0.00 | 0.02 | 0.01 | 0.00 |
| 5.45 | 0.00 | 0.00 | 0.02 | 0.02 | 0.01 |
| 4.46 | 0.00 | 0.00 | 0.02 | 0.01 | 0.00 |
| 3.27 | 0.01 | 0.01 | 0.02 | 0.01 | 0.01 |
| 2.73 | 0.01 | 0.01 | 0.02 | 0.01 | 0.01 |
| 2.34 | 0.02 | 0.01 | 0.02 | 0.01 | 0.01 |
| 1.96 | 0.02 | 0.02 | 0.02 | 0.02 | 0.01 |
| 1.75 | 0.02 | 0.02 | 0.02 | 0.03 | 0.02 |
| 1.40 | 0.05 | 0.03 | 0.02 | 0.02 | 0.02 |
| 1.23 | 0.03 | 0.03 | 0.02 | 0.03 | 0.03 |
| 1.09 | 0.04 | 0.03 | 0.03 | 0.04 | 0.03 |
| 1.00 | 0.04 | 0.03 | 0.02 | 0.03 | 0.03 |

GEP models for all log-transformed respirations types time series, before back-transformation.

$$\log\left(R_{eco}\right) = \frac{GPP_s}{T_{-10}} + \log\left(\log\left(T_{-10}\right)\right) \tag{1}$$

$$\log\left(R_{above}\right) = 0.1T_{-10} + 0.4\log\left(0.8\sqrt{SWC}\right) \tag{2}$$

$$\log\left(R_{soil}\right) = 1.2T_{-10}^{0.4} + 1.3SWC - 3.1 \tag{3}$$

$$\log\left(R_{root}\right) = 0.9\frac{1.2GPP_s - 8.1}{T_{-10}} \tag{4}$$

$$\log\left(R_{myc}\right) = 1.1\log\left(1.7T_{-10}\right) + 1.2T_{-10}^{SWC} - 7.4 \tag{5}$$

$$\log\left(R_{soil_a}\right) = 1.2T_{-10}^{0.5} + 2.5SWC - 4.9 \tag{6}$$

$$\log\left(R_{soil_h}\right) = -0.3 + 0.6\frac{1.1GPP_s - 3.6}{T_{-10}} \tag{7}$$

Figure 1 in supplemental material illustrates the change in the shape of the PDF estimated for each respiration type after log-transforming. For all time series, the skewness is visibly is reduced.

**Table 2.** Standard error of the MEF at validation values for all MLM for different SNR values when the MEF values are computed against the clear data.

| SNR | GEP | KRR | RF | SVM | ANN |
|------|------|------|------|------|------|
| 9.82 | 3e-07 | 4e-05 | 2e-02 | 4e-03 | 4e-03 |
| 8.18 | 3e-07 | 6e-05 | 2e-02 | 2e-02 | 2e-03 |
| 7.01 | 3e-07 | 4e-05 | 2e-02 | 1e-02 | 2e-03 |
| 6.14 | 2e-06 | 7e-05 | 2e-02 | 2e-02 | 2e-03 |
| 5.45 | 2e-06 | 1e-04 | 2e-02 | 2e-02 | 4e-03 |
| 4.46 | 6e-06 | 1e-04 | 2e-02 | 2e-02 | 2e-03 |
| 3.27 | 9e-06 | 2e-03 | 2e-02 | 1e-02 | 3e-03 |
| 2.73 | 4e-05 | 4e-04 | 2e-02 | 1e-02 | 6e-03 |
| 2.34 | 4e-05 | 6e-04 | 2e-02 | 9e-03 | 3e-03 |
| 1.96 | 8e-05 | 1e-03 | 2e-02 | 1e-02 | 3e-03 |
| 1.75 | 2e-04 | 8e-04 | 1e-02 | 1e-02 | 5e-03 |
| 1.40 | 8e-04 | 1e-03 | 1e-02 | 2e-02 | 5e-03 |
| 1.23 | 1e-04 | 2e-03 | 1e-02 | 2e-02 | 4e-03 |
| 1.09 | 4e-03 | 3e-03 | 1e-02 | 2e-02 | 5e-03 |
| 1.00 | 7e-04 | 3e-03 | 1e-02 | 5e-02 | 6e-03 |

From Fig. 5 and 6 is worth mentioning the apparent correlation, although weak in terms of $R^2$ value, of the $R_{myc}$ residuals with $GPP_s$, even when this was not chosen as a driver, indicating that the relation was not strong enough for an explicit model inclusion but it could show a dependency to a driver for which $GPP_s$ acts as a proxy such as phenology, or substrate availability. Such weak correlations are present as well between $R_{soil}$ and $R_{soil_h}$ residuals and $T_{air}$.

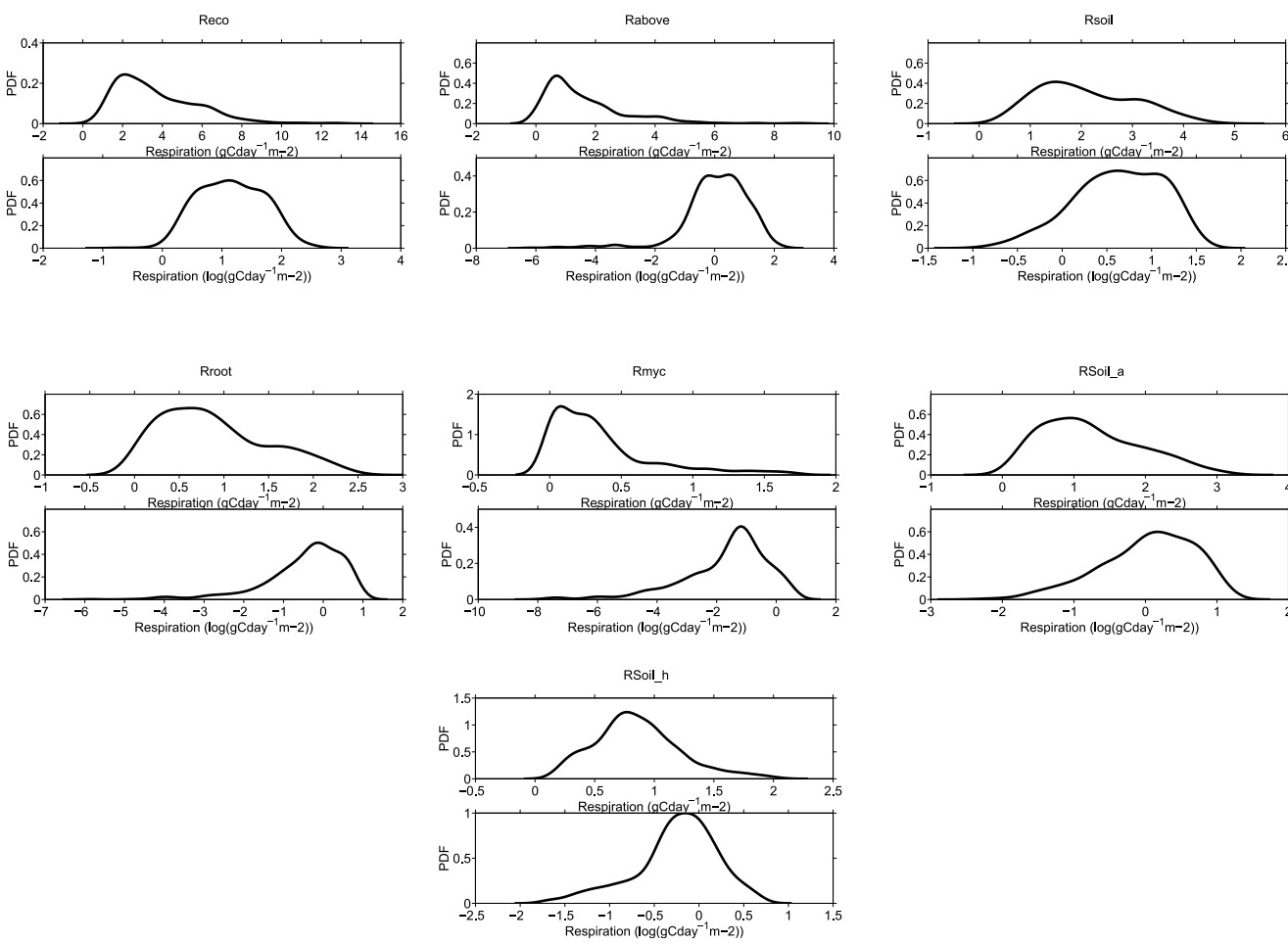

**Figure 1. Change in estimated density function of observations before and after log-transforming for all studied respiration types.**

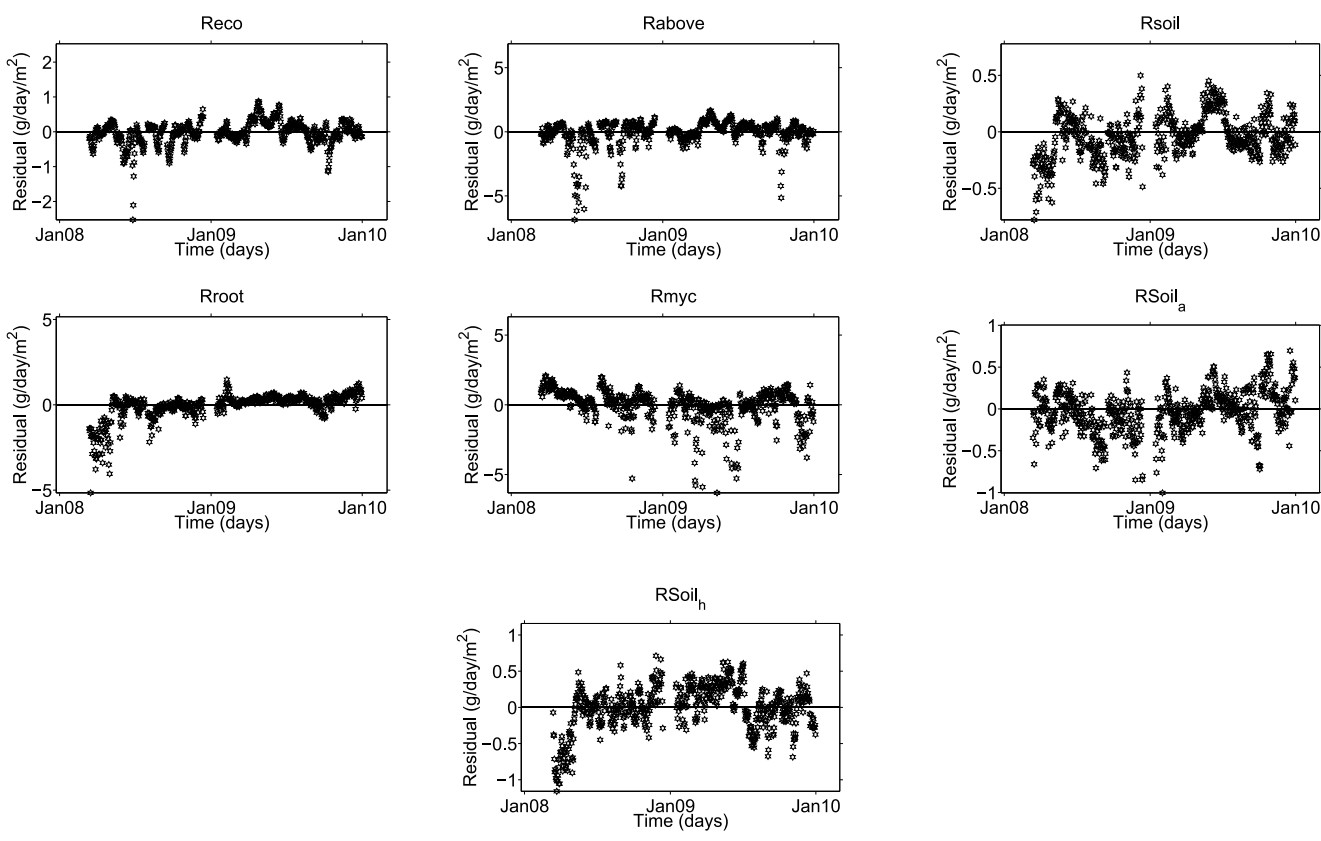

**Figure 2. Residuals computed for the GEP models against the log-transformed targets before back-transformation.**

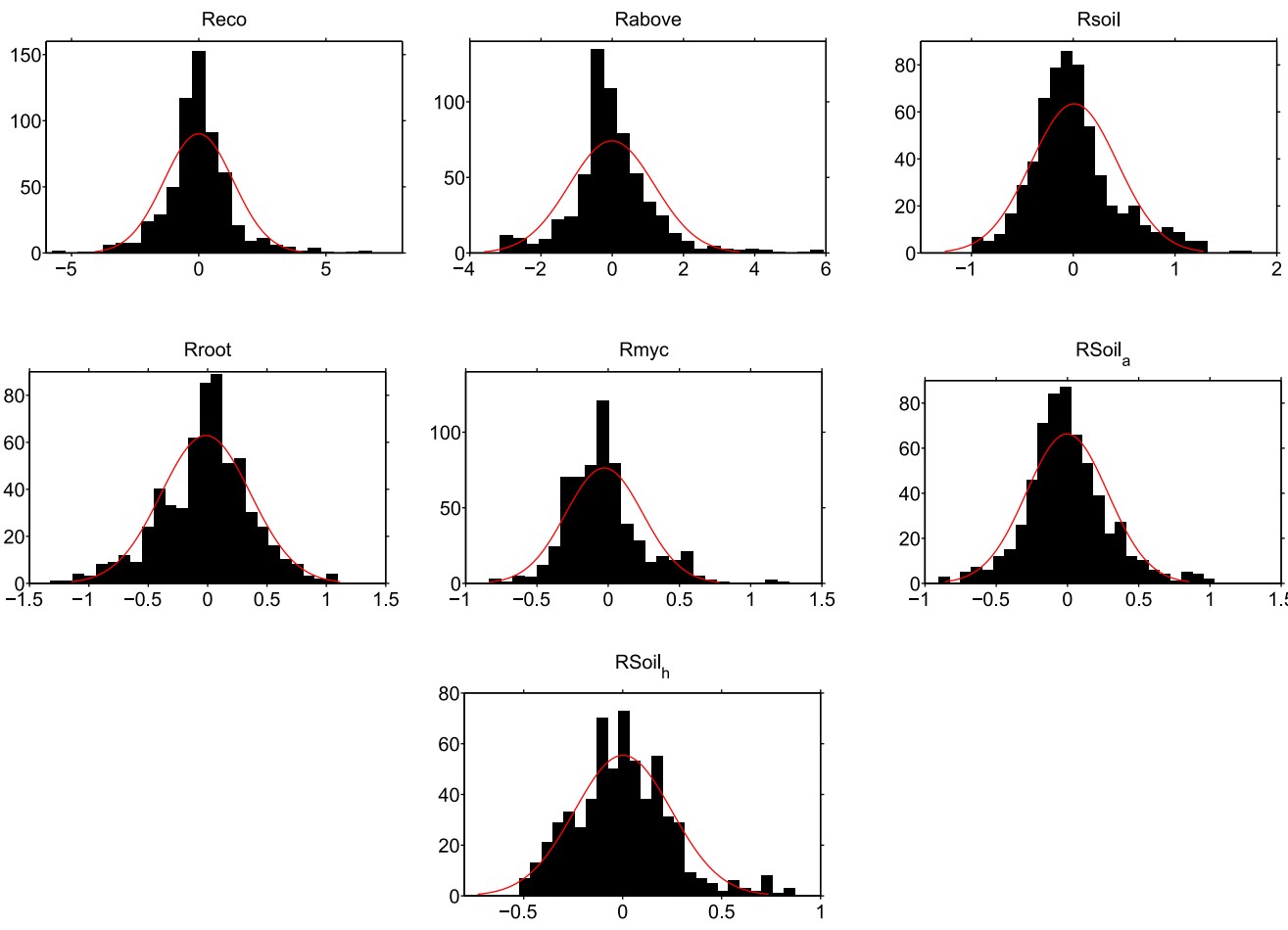

**Figure 3. Distributions of the residuals after smear bias correction computed for the GEP models after training on log-transformed data.**

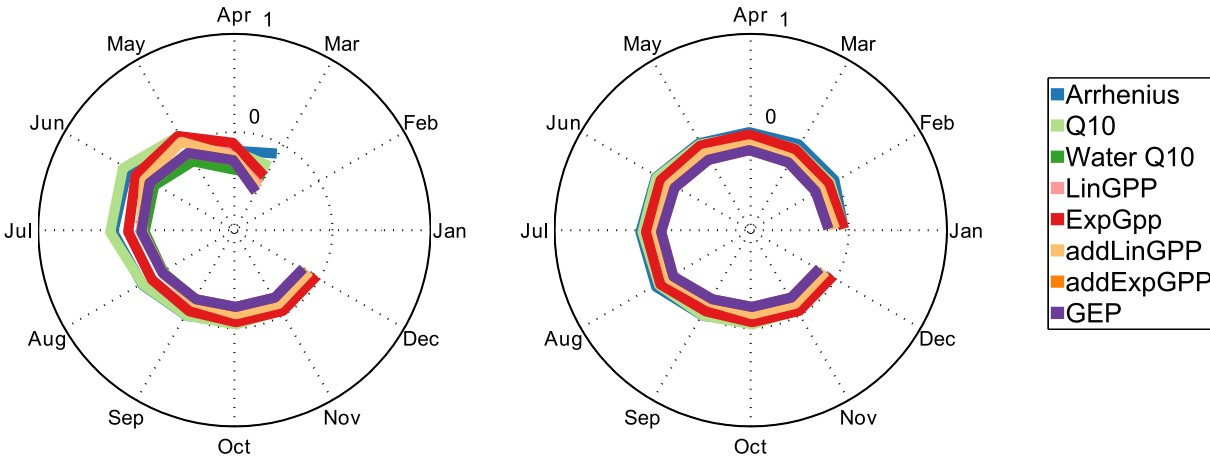

**Figure 4. Monthly averaged error values for some literature models for and the GEP generated model** for daily soil $CO_2$ efflux in the two studied years. The centre of the plots is -1. The scale of the fluxes is given in gC/$m^2$/day.

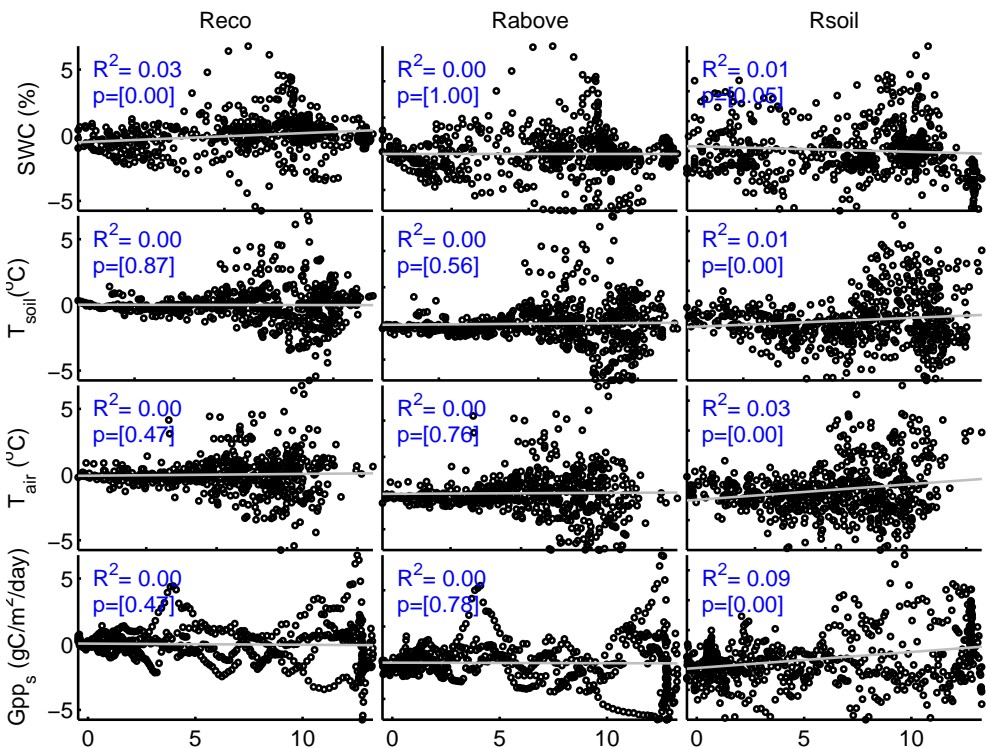

**Figure 5. Candidate driver linear correlations with residuals** computed after bias corrected transformation of the GEP models from runs with settings given in Tab 1 for $R_{eco}$, $R_{above}$ and $R_{soil}$. The drivers are on the X axis and the residuals on the Y axis. The candidate driver is given as title of each row and the type of respiration is given as title of the column.

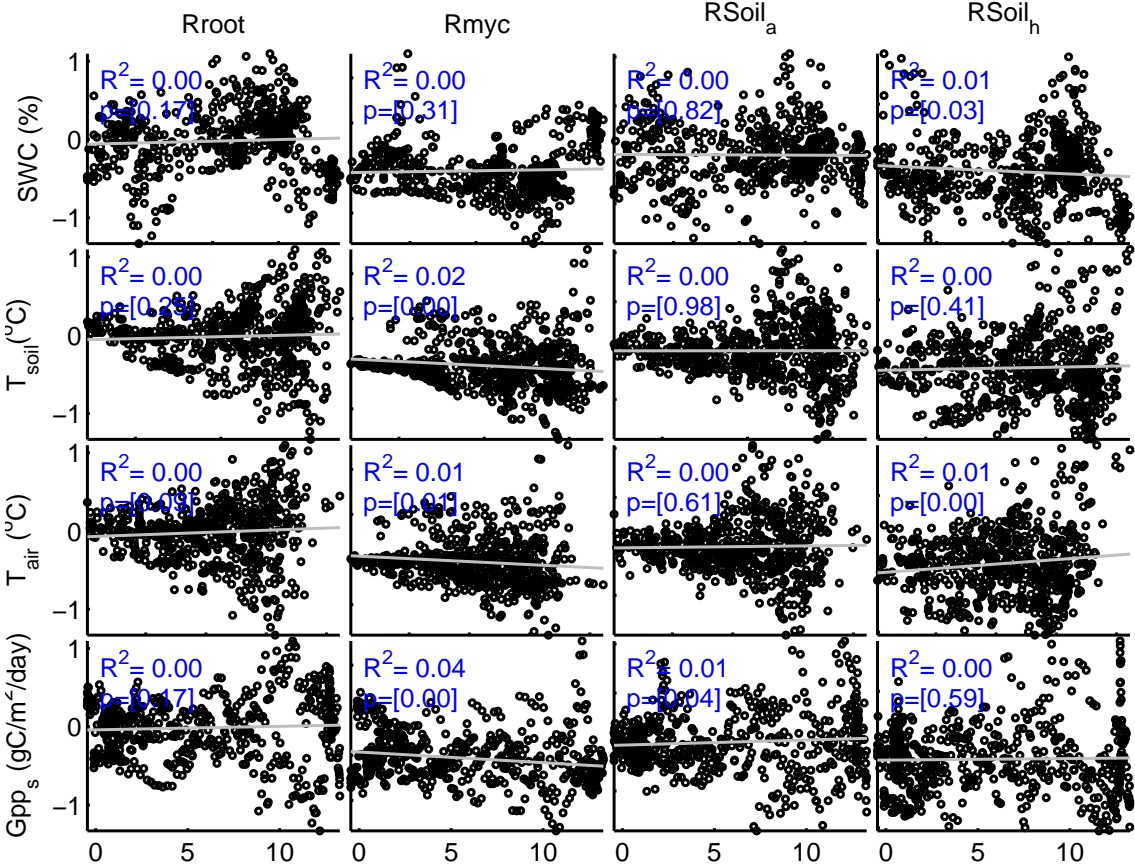

**Figure 6. Candidate driver linear correlations with residuals** computed after bias corrected transformation of the GEP models from runs with settings given in Tab 1 for $R_{root}$, $R_{myc}$, $R_{soil_a}$ and $R_{soil_h}$. The drivers are on the X axis and the residuals on the Y axis. The candidate driver is given as title of each row and the type of respiration is given as title of the column.