# Peer review of "Reverse engineering model structures for soil and ecosystem respiration: the potential of gene expression programming"

_Geoscientific Model Development, 2016_

## Referee Comment (RC1) · Anonymous Referee #1 · 22 Dec 2016

The manuscript proposes to automatically derive model structures using Gene Expression Programming (GEP) introduced by Ferreira (2001). The authors apply GEP to different components of terrestrial $CO_2$ fluxes measured in an 80 year old deciduous oak plantation in the Alice Holt forest in SE England. The goal is to compare automatically derived model structures with predictions by other machine learning methods and from other published models of ecosystem respiration.

The paper is in the scope of the journal and the topic could be interesting for a broad audience of geoscientists.

In the present form, I cannot recommend publishing and ask the authors to thoroughly review their manuscript taking the below mentioned points into account. Additionally,

the manuscript would benefit from a proofreading by a native speaker.

Major comments

1. The goals stated in the introduction are scattered over page 3 (ll. 3–4, ll. 23–25, ll. 28–30). Please state them clearly at the end of the introduction.

2. GEP is the key part of the manuscript. It is not a standard modelling framework and needs a clear introduction. In the present form, Section 2.1 is difficult to understand for someone not familiar with GEP. Please define clearly what is a gene, a chromosome and an expression tree and how they are related. Use examples for illustration. The original paper by Ferreira (2001) is written for a broad readership and can serve as an example. How are the mathematical statements coded in chromosomes evaluated to generate predictions?

3. The use of the fitness measures is inconsistent throughout the manuscript. In section 2.2 you derive a composite fitness measure CEM and state that this is your final normalized form of the fitness function (eq. 2.3). However, later in the results you report MEF or MEF+NP (that was never property introduced). Explain clearly which function was used to measure the fitness. Also p. 8 l. 4–5 shows that CEM is apparently not your final fitness function.

4. What were the functions that were coded in GEP and could thus form algebraic expressions? How did you chose them?

5. Section 3.1.1: You state the the machine learning methods (Artificial Neural Networks, Support Vector Machines, Random Forests and Kernel Ridge Regression) were used without tuning the hyperparameters. I have a serious objection here. While some of the hyperparameters could be safely set to default values, others have to be tuned and do affect the performance of those models (e.g. the

cost parameter of Support Vector Machines). I recommend that you consult the technical literature here and tune hyperparameters for a fair comparison. A good point to start is the book by Kuhn and Johnson (2013).

6. Which predictors did you use for the machine learning methods on the artificial data?

7. p. 9 l. 29–32 You state that you log-transformed the fluxes before modelling and back-transformed the model structures. Did you also back-transform the predictions? At least in standard regression, back-transformations need particular attention. When back-transforming from the log transformation, the variance of the residuals has to be considered in order to avoid a bias. Please explain what and how you back-transformed. How did you take care of a possible bias?

8. Fig 8 shows a lot of dynamics in residuals from the GEP approach. Because you are dealing with time series, reporting MEF only is not satisfactory. A more in depth comparison of the different models at different time scales is appropriate (e.g. Mahecha et al., 2010). Which temporal patterns can be well reproduced by the different models?

9. From Fig 10 we learn that the machine learning algorithms performed better than GEP. In Section 5.2 you state that GEP underestimates high fluxes as do the published semi-empirical models. So what is the advantage of using a GEP approach? What can we learn from it? I suggest that you restructure your discussion such that this aspect becomes really clear. In the present form Section 5.2 is somehow lost.

Detailed comments

- p. 3 l. 14 Explain briefly symbolic regression here and in more details in the method section (p. 4 ll. 9ff).

- p. 4 l. 14–17 You state that the "variables and functions are subsequently mapped to a set of characters", then that the "mapping process generates sets of strings. . . " And then in the next sentence "the mapped letters are randomly combined . . . ". This is confusing. State clearly what is the alphabet used to map functions and variables. They cannot be randomly combined: a binary function has to have two inputs, for example, and this is taken care of in the coding sequence. The initial chromosomes are generated randomly, however, the genes must be valid mathematical expressions.

- p. 4 l. 32 explain individual.

- p. 5 l. 1 How is the hyperparameter tuned?

- p. 5 l. 8–9 How is the population diversity related to stochastic bias?

- p. 6 l. 2 and eq. 2.2 inconsistent names: SE or S[P]?

- Give more details on the calculation of the permutation entropy (Bandt and Pompe, 2002). A reader not familiar with the method should be able to understand what you calculated.

- Eq. 2.3. I don't understand the last term in your derivation of CEM. Why $1 - SE$? The permutation entropy varies between 0 and $log(n!)$, $n$ being the order of permutation ($n = 4$ in your case). Did you normalise SE by its maximum?

- Is CEM maximized or minimized?

- p. 6 l. 22 Why are model parameters constant values? This term for an entity being optimized is confusing.

- p. 7 l. 26 and Tab1: You never explained head and tail of genes.

[Figure]

- p. 9 l. 25 Explain briefly how the Singular Spectrum Analysis works and give references to the original publications (Broomhead and King, 1986, for example).

- p. 10 l. 1–2 I don't understand how your split you data in training and test data sets. According to p. 8 l. 21 you have two years of hourly observations. So what are the 500 target time steps and why are there 613 time steps in total? How did you calculate the subsets?

- p. 12 l. 20–22 What do you mean by a "component of $R_{eco}$ not seen in the training procedure"? Which components were not modelled?

- p. 14 l. 10 Which water reservoir do you refer to? Soil water? Then reservoir is misleading.

- p. 16 l. 13 You state that GEP is not prone to overfitting. How did you analyse this?

- What are the error bars in Fig3(a), (b) and fig4 (c)?

- Fig3(c) is not necessary.

- Fig12 is never discussed in the text.

**References**

Bandt C and Pompe B. 2002. Permutation entropy: a natural complexity measure for time series. *Physical review letters*, **88**(17): 174102.

Broomhead D and King GP. 1986. Extracting qualitative dynamics from experimental data. *Physica D: Nonlinear Phenomena*, **20**(2-3): 217–236.

Ferreira C. 2001. Gene expression programming: A new adaptive algorithm for solving problems. *Complex Systems*, **13**(2): 87–129.

Kuhn M and Johnson K. 2013. *Applied predictive modeling*. Springer.

Mahecha MD, Reichstein M, Jung M, Seneviratne SI, Zaehle S, Beer C, Braakhekke MC, Carvalhais N, Lange H, Le Maire G, and Moors E. 2010. Comparing observations and process-based simulations of biosphere-atmosphere exchanges on multiple timescales. *J. Geophys. Res.*, **115**(G2): G02003–.

——————————————————————

---

## Referee Comment (RC2) · Anonymous Referee #2 · 24 Dec 2016

Review of "Reverse engineering model structures for soil and ecosystem respiration: the potential of gene expression programming"

In this manuscript Ilie et al. explore the use of "gene expression programming" (GEP) to select empirical models for soil and ecosystem respiration. The authors make a case that GEP is a technique for reverse engineering model structures by elucidating underlying mechanisms, rather than depending on hypothesis-driven experiments to identify these mechanisms.

I have several concerns about the conceptual framework the authors used to present GEP. I am convinced that GEP is an interesting and worthwhile approach to automate model selection. However, I think it is over-reaching to suggest that GEP can

'reverse engineer' model development. It seems to me that the value of GEP is simply to automate the process of exploring a large number of regression models. I am not convinced that GEP reorganizes the model development process, because regression already is often the first step in model development. Further, I find that the claim that GEP minimizes human influence and perception bias to be strong, as the authors seemingly arbitrarily select the driving variables for the model, regardless of how the model's functional form is derived. From other work we know that selecting a single soil temperature at 5 cm soil depth can give a very different model from selecting a temperature from 15 cm soil depth (Graf et al., Biogeosciences doi:10.5194/bg-5-1175-2008). Similarly selecting to use VWC rather than a parameter like matric potential could be the difference between being able to predict rapid increases in flux with rainfall and not.

In the end, the functions selected by GEP suffer from the same problems as previously-used formulae shown in Table 2. All of these functions tend to underestimate large fluxes ("hot spots" and "hot moments"). While the form of the functions may hold-up from training datasets to prediction datasets, the specific parameterizations often do not. I believe the authors have done a good job discussing limitations of GEP, and empirical approaches in general, in section 5.1.1. We know biogeochemical fluxes integrate multiple pools, reservoir dynamics and lags, and these are difficult to detect using semi-empirical models. The largest gains recently in representing soil respiration have come from simulating enzyme kinetics and solute diffusion (e.g. DAMM model) as well as simulating microbial growth dynamics. These advances have come from implementing expert knowledge, not from expediting regression model selection.

Overall I would recommend that this manuscript be rejected in the current form, and the authors re-evaluate the presentation of the GEP method both in terms of creating certainty within the biogeosciences community that the approach is effective and accessible, as well as readily applicable to field data as was demonstrated with the data from Alice Holt. As was mentioned, I believe the GEP method has considerable potential, but as the manuscript is currently written my concern is that it will pass unnoticed

by the community as a whole due to poor accessibility rather than scientific merit.

General Comments: I do not agree with Figure 1, that model development starts with expert knowledge. Expert knowledge does not come about on its own, but comes from observations, and regressions are critical to making sense of observations. By helping to identify which variables among a large number of potential explanatory variables correlate to a phenomenon, regression-type analyses lead to the second step in the scientific process: manipulative experiments to confirm hypothesized cause-and-effect relationships. Demonstrating cause-and-effect relationships limits the number of processes that need to be represented in models. I am not convinced that GEP provides a short-cut to this process.

Section 3.1 and 4.1, which outline artificial experiments with the GEP method could be strengthened considerably if the authors were to us a simple, mechanistic model of soil or ecosystem respiration rather than a seemingly random set of algebraic expressions. Using such a respiration model would allow the authors to attempt to recover the model basis functions and, if successful, enhance the reader's confidence with respect to the data from the site at Alice Holt.

I am concerned about the evaluations of GEP presented in Figs. 3 and 4. Fig 3 compares alternate machine learning techniques by comparing the MEF of the final model selected by each approach. It seems to me also important to compare the actual model structures, not just the fitness score. Did all the techniques recover the original models? If not, is variation in the MEF meaningful?

Figure 4c suggests that GEP was only able to recover about 30-55% of the correct number of parameters. If so, it seems GEP did NOT do a good job of recovering the original models.

Another major concern is the exercise shown in figure 7. The authors have examined whether summing predicted component fluxes gives predicted total fluxes that resemble observations. This is an interesting idea, but ultimately not that useful for two reasons: 1) The observed fluxes were not independently measured, e.g. Rauto was not measured independently, but was calculated by measuring the total flux (Rsoil) minus RH. I think you want to test whether all the variability simulated for the components can explain the variability observed for the total flux, but you don't have a measure of the component fluxes independent from the total flux. 2) We would like to see that the predictions for total flux are no worse than the predictions for the component fluxes. But in several cases the prediction for component fluxes are pretty poor. E.g. Predictions for RECO won't turn out any better than predictions for Rabove, which themselves were poor. That's not so interesting.

The manuscript is figure heavy, consider condensing figures or removing. For example can Figures 5 and 9 be combined in an effective way? Are there other figures that may be unnecessary to the reader if they were described in the text or in a table?

Specific comments: Abstract is long, introduces a lot of terminology. Consider distilling to the most important take-homes, and make more approachable for a general audience. p.3 l. 8. The rationale for reordering should also be to try more options, things that people might miss p. 3. L. 30. Why would we expect the functions to be portable across scales? Provide an ecological justification, otherwise this is not an interesting or useful exercise. p. 3. L. 22-35. When reading initially I found it difficult to understand what hypotheses the authors were testing. I think all of this information is there but needs to be re-organized to make it stand out to the reader. p.4 ll. 5. No need to introduce the conclusions. Consider shortening this to reduce repetition. 2.1 This section was not clearly written, I suggest more careful editing by co-authors. Please avoid including extra words in parantheses, they add complexity without clarity. p.4 ll.15. Is the process of mapping operations to strings relevant to model fitting? I don't think so. Either this is excessive detail about the internal workings of GEP, or you need to explain how this is relevant. p. 4. L. 20, what do you mean by "solution" The final selected model? Or the respiration predicted by that model? "Genes" and "chromosomes" should be presented in quotations initially. p. 4 l. 30 I think you

can shorten this paragraph to one sentence, simply state that in each generation, the best variants of a chromosome are determined by a fitness function described below. p. 4 l. 32, what is an individual? Do chromosomes make up individuals? p. 5, l. 1 What is a hyper-parameter? Again, please try to avoid parenthetical phrases in this paragraph. p. 5, ll. 12 "upon request" rather than "on demand". p. 5, l. 11-14 most of this information doesn't appear useful, for example, does it actually matter that the cluster had 51 nodes? If someone ran it on a cluster with 12 nodes would it also work but be slower? Either explain the relevance of these details or remove them. p. 5, ll. 31 Consider omitting "derived from information-theoretic considerations". p. 6, ll. 20-25. I didn't understand the reason for this additional optimization. This sounds very much like ordinary regression model selection; does this undermine the unique value of GEP? p. 6, ll. 27 Scaling noise with signal amplitude: This is good to include! This has been shown for soil respiration too (Lavoie et al. 2015, JGR-Biogeosciences, doi: 10.1002/2014JG002773) Section 3.2.1 The first two paragraphs are repetitive in describing computation of GPP.Consider omitting or shortening the section on soil flux measurements, since these methods were reported previously. Section 3.2.4 This paragraph can be removed to shorten. Figure 3c, consider omitting. It is repetitive, and the manuscript already has a large number of figures. p.12, l. 7 Sentence starting "We find that the global modelling performance…" Please reword, I don't understand this statement. Figure 12, is there a reason that this is presented in a polar plot? It seems on first glance that it could equally be presented as a 4-pane set of cartesian time series plots.

––––––––––––––––––––––––––––––––

---

## Author Comment (AC1) · 1 Mar 2017

**1   Response to Reviewer 1**

In the following, we denote comments by the reviewer in **bold** and our own reponses in standard fonts.

**The manuscript proposes to automatically derive model structures using Gene Expression Programming (GEP) introduced by Ferreira (2001). The authors apply GEP to different components of terrestrial CO2 fluxes measured in an 80 year old deciduous oak plantation in the Alice Holt forest in SE England. The goal**

**is to compare automatically derived model structures with predictions by other machine learning methods and from other published models of ecosystem respiration. The paper is in the scope of the journal and the topic could be interesting for a broad audience of geoscientists. In the present form, I cannot recommend publishing and ask the authors to thoroughly review their manuscript taking the below mentioned points into account. Additionally, the manuscript would benefit from a proofreading by a native speaker.**

We would like to thank the reviewer for the evaluation and detailed comments on our manuscript. We further provide responses for the posed questions and details on how we intend to revise the manuscript. Please note that our UK based co-authors had revised the paper, and will again be involved in the submission of the revised manuscript.

**Major comments**

1. **The goals stated in the introduction are scattered over page 3 (ll. 3–4, ll. 23–25, ll. 28–30). Please state them clearly at the end of the introduction.**

   • The section can and will be re-organized as suggested by the reviewer in the revised manuscript.

2. **GEP is the key part of the manuscript. It is not a standard modelling framework and needs a clear introduction. In the present form, Section 2.1 is difficult to understand for someone not familiar with GEP. Please define clearly what is a gene, a chromosome and an expression tree and how they are related. Use examples for illustration. The original paper by Ferreira (2001) is written for a broad readership and can serve as an example. How are the mathematical statements coded in chromosomes evaluated to generate predictions?**

- Thank you four pointing this out here. We will include a figure explaining the process of mapping and evaluation of strings to mathematical functions in the revised version of the manuscript. We also work on describing more carefully what we understand here as "gene", "chromosome" and an "expression tree". We agree that this is absolutely key to the readers.

3. **The use of the fitness measures is inconsistent throughout the manuscript. In section 2.2 you derive a composite fitness measure CEM and state that this is your final normalized form of the fitness function (eq. 2.3). However, later in the results you report MEF or MEF+NP (that was never property introduced). Explain clearly which function was used to measure the fitness. Also p. 8 l. 4–5 shows that CEM is apparently not your final fitness function.**

   - We apologize that we have not been sufficiently clear in our descriptions: CEM=MEF+NP+SE (modelling efficiency +number of parameters+ signal complexity measure) is the final fitness function used for optimizing the solutions for all GEP results presented in this paper. The MEF values are reported for quantifying the model-data misfit which is more natural to "read". More explanations on MEF+NP will be added to the revised manuscript as well. This function is a fitness function similar to CEM, but where the entropy component is missing. This function was introduced in the manuscript in order to better illustrate the effect of each fitness function component for the final GEP solutions performance.

4. **What were the functions that were coded in GEP and could thus form algebraic expressions? How did you chose them?**

   - Usually in genetic programming type of approaches, the identification of input functions depends on the type of problem which we try to solve. If we tackle symbolic regressions, as is the case here, most often a set of primitive

functions is proposed and sufficient, such as addition, multiplication, exponential and so on. More complex functions could increase model complexity too much and risk overfitting. We will add a more detailed explanation in the revised manuscript.

5. **Section 3.1.1: You state the the machine learning methods (Artificial Neural Networks, Support Vector Machines, Random Forests and Kernel Ridge Regression) were used without tuning the hyperparameters. I have a serious objection here. While some of the hyperparameters could be safely set to default values, others have to be tuned and do affect the performance of those models (e.g. the C2 cost parameter of Support Vector Machines). I recommend that you consult the technical literature here and tune hyperparameters for a fair comparison. A good point to start is the book by Kuhn and Johnson (2013).**

- *W*e are sorry for the confusion here: we wrote that "All the runs were performed with default settings" e.g. regarding the choice of their Kernels. But we did, of course allow the hyper parameters to vary and adjusted them in a cross-validation approach as described in Camps-Valls2012.
  The only approach run with default settings was the RF approach from the matlab statistics toolbox implementation.
  The paragraph should say (p8 l9):
  "The toolboxes and settings used for generating the predictions of the ANN and KRR methods are described by Tramontana2016 and found in the "simpleR" regression toolbox Lazaro-Gredilla2014, the predictions of the SVM were obtained by using the "LIBSVM" library Chang2011 from the "simpleR" regression toolbox where the regularization term, the insensitivity tube (tolerated error) and a kernel length scale are automatically adjusted. Lastly, the RF predictions were given by the Matlab statistics toolbox implementation running with default settings. "

[Figure]

Was corrected in the manuscript p8 l9-12.

6. **Which predictors did you use for the machine learning methods on the artificial data?**

   - Thank you for pointing this aspect out. All the machine learning methods (GEP, KRR, ANN, SVM and RF) learn based on the same input data set for all artificial problems, which contains 3 candidate variables ($x_1$, $x_2$ and $x_3$), which means that all methods are allowed to perform a feature selection as well. We apologize that this was no made clear in the manuscript but we have now corrected that p7 l25-26.

7. **p. 9 l. 29–32 You state that you log-transformed the fluxes before modelling and back-transformed the model structures. Did you also back-transform the predictions? At least in standard regression, back-transformations need particular attention. When back-transforming from the log transformation, the variance of the residuals has to be considered in order to avoid a bias. Please explain what and how you back-transformed. How did you take care of a possible bias?**

   - For the GEP solutions, we trained on log-transformed target data. That gave us a set of solutions. But of course, in order to obtain the initial fluxes an exponential function was applied to these solutions. From the exponential functions we obtained predictions which are further compared with the original target data and MEF values were reported. So, yes - we back-transformed the resulting structures.
   - For the remaining machine learning approaches (ANN, SVM, RF and KRR) the exponential is applied directly to the predictions obtained after learning from the log-transformed target and the resulting predicted fluxes are compared with the original target by means of MEF.

- We don't exactly understand the issue of the bias - it would actually matter during the optimization as the cost-function deals with the log-transformed data. But after back transforming, the data are in original space and the evaluation with the MEF should be fine. This means also that the model selection should be unbiassed.

8. **Fig 8 shows a lot of dynamics in residuals from the GEP approach. Because you are dealing with time series, reporting MEF only is not satisfactory. A more in depth comparison of the different models at different time scales is appropriate (e.g. Mahecha et al., 2010). Which temporal patterns can be well reproduced by the different models?**

   - We agree that MEF is a bit superficial. However, Figure 12 reveals that model-data miss-match is not only an issue of a certain fast time scale, but clearly also occurs on seasonal time scales. The Mahecha2010 approach is very useful if we would be able to additionally deal with e.g. trends etc. But for this kind of analysis the time-series are simply too short.

9. **From Fig 10 we learn that the machine learning algorithms performed better than GEP. In Section 5.2 you state that GEP underestimates high fluxes as do the published semi-empirical models. So what is the advantage of using a GEP approach? What can we learn from it? I suggest that you restructure your discussion such that this aspect becomes really clear. In the present form Section 5.2 is somehow lost.**

   - Thank you for pointing this out. Indeed, this discussion is at the heart of our philosophical approach: We argue that if GEP identifies structurally very different model that, however, yield equivalent model performance, it puts at question the validity of the conventional semi-empirical models. GEP models reveal that certain dynamics that are typically unconsidered in approaches of this kind, for instance the exponential influence of SWC to res-
piration components or the seasonal influence of GPP. This section of the
discussion will be restructured in the revised manuscript for increased clari-
fication of where we see the added value of such an approach.

**Detailed comments**

- **p. 3 l. 14 Explain briefly symbolic regression here and in more details in
the method section (p. 4 ll. 9ff). C3**

  A symbolic regression is a type of regression where not only the parameters of
  a known (linear) function are optimized based on data, but where the functional
  form itself is also constructed based on data as a combination of basic linear and
  non-linear mathematical functions. Further expanded in the method section.

- **p. 4 l. 14–17 You state that the "variables and functions are subsequently
  mapped to a set of characters", then that the "mapping process generates
  sets of strings. . . " And then in the next sentence "the mapped letters
  are randomly combined . . . ". This is confusing. State clearly what is
  the alphabet used to map functions and variables. They cannot be ran-
  domly combined: a binary function has to have two inputs, for example,
  and this is taken care of in the coding sequence. The initial chromosomes
  are generated randomly, however, the genes must be valid mathematical
  expressions.**

  The input variables and functions are indeed mapped to characters that are com-
  bined into strings which encode the mathematical expressions. The validity of
  encoded mathematical expression is insured by the internal translation language
  and by the equation: tail=head*2+1. Thus although each of the sections, head
  and tail are generated based on random selection from the input characters sets
  (functions+variables sets for head and variables for tail), there are still rules that

insure validity of mathematical expressions (except for cases where a solution can only be deemed invalid by evaluating the expression, such as division by 0, etc)

- **p. 4 l. 32 explain individual.**

  The individual is a component of the evolution population which encodes a specific mathematical expression. It is the same as chromosome. Added better definition in glossary.

- **p. 5 l. 1 How is the hyper-parameter tuned?**

  The hyper-parameter has either some commonly used default values in the community, especially for the genetic operators rations, or some values that have been empirically established with experience, depending on the problem we are looking at.

- **p. 5 l. 8–9 How is the population diversity related to stochastic bias?**

  Once diversity is insured in the evolution population, we can be more confident that a certain solution does not appear just by chance, as it would have to be good enough to beat a larger pool of solutions.

- **p. 6 l. 2 and eq. 2.2 inconsistent names: SE or S[P]?**

  SE is the name we use for the Shannon entropy. S[P] is changed as well to SE in the manuscript.

- **Give more details on the calculation of the permutation entropy (Bandt and Pompe, 2002). A reader not familiar with the method should be able to understand what you calculated.**

  In short, the calculation of an entropy as a measure for randomness from a time series (e.g. Shannon's entropy) requires to determine a probability distribution

that underlies the time series (or dynamical system), which is usually done by a partitioning step (also called phase space reconstruction in other contexts). This is a fundamental step in the methodology, and various methods have been used to arrive at this probability distribution, for instance frequency or histogram-based measures, procedures based on amplitude statistics, or symbolic dynamics (see e.g Kowalski et al 2011 for an overview). In recent years, the Bandt Pompe approach has become popular, because it directly takes sequences in time into account: The technique hence divides the time series into ordinal sequences (i.e. ordinal patterns, or symbolic sequences), and then computes entropy measures directly from the probability distribution of these ordinal patterns Bandt2002. This approach has a number of advantages, namely that it is robust to noise (no sensitivity to numeric outliers) and to trends or drift in the data, it is an (almost) non-parametric method and no prior assumptions about the data are needed (the only parameter that has to be specified is the embedding dimension, i.e. window length), and allows to disentangle various possible states of the system that are then encoded in the probability distribution (see e.g. Zanin2012 for a review of the method and applications). We will describe this method in more detail, and give a few examples of its application in the revised manuscript.

- **Eq. 2.3. I don't understand the last term in your derivation of CEM. Why $1 - SE$? The permutation entropy varies between 0 and log(n!), n being the order of permutation (n = 4 in your case). Did you normalise SE by its maximum?**

  SE is indeed normalized by its maximum; hence SE varies between 0 and 1, where 1 indicates no correlated structure in the residuals. Furthermore, the best CEM value can take, and towards which the optimized values tend to is 0.

- **Is CEM maximized or minimized?**

[Figure]

Throughout the entire paper, the optimization is done by minimization of the fitness function value.

- **p. 6 l. 22 Why are model parameters constant values? This term for an entity being optimized is confusing.**

GEP as a method does not offer a specific optimization of parameters, as it evolves entire mathematical formulations. So until there is a special treatment in terms of optimisation for the parameters, they are considered constants. Once a final solution is reached, a specific optimization algorithm is used for

- **p. 7 l. 26 and Tab1: You never explained head and tail of genes.**

We apologise for the slip. Added to glossary.

- **p. 9 l. 25 Explain briefly how the Singular Spectrum Analysis works and give references to the original publications (Broomhead and King, 1986, for example).**

The SSA method is a very useful tool used mainly in time series analysis with the purpose of decomposing an original time series into the sum of its components, such as trends, seasonality and high frequency components. More details and the references are added to the revised manuscript.

- **p. 10 l. 1–2 I don't understand how your split you data in training and test data sets. According to p. 8 l. 21 you have two years of hourly observations. So what are the 500 target time steps and why are there 613 time steps in total? How did you calculate the subsets?**

Thank you for pointing this aspect out. It seems that we have not been clear enough in the description. Data is available with hourly resolution, however, we use daily means for model constructions. So for two years, we should have 732 data points, but after filtering we are left with a gapped set of 613 observations.

Those 613 d.p. are split into two sets of 500 and 113 d.p 50 times. For each of this split we then learn a model and the best over-all at validation is finally selected and presented in the results section. Section is revised for clarity.

- **p. 12 l. 20–22 What do you mean by a "component of Reco not seen in the training procedure"? Which components were not modelled?**

Each component was separately modelled and a solution is built with GEP. Then, the parameters of each of these solutions are re-calibrated using CMA-ES for the rest of the components for a fair comparison of modelling capacity.

- **p. 14 l. 10 Which water reservoir do you refer to? Soil water? Then reservoir is misleading.**

Indeed we refer to soil water. We apologize for the confusion and water reservoir has been changed to soil water in throughout the revised manuscript.

- **p. 16 l. 13 You state that GEP is not prone to overfitting. How did you analyse this?**

This was concluded for the results of the increase of signal to noise ratio exercise, as the MEF values of the solutions reconstructed when compared to original, noise free data do not change significantly with addition of noise.

- What are the error bars in Fig3(a), (b) and fig4 (c)? The error bars are not visible enough at the scale of the plot. A table with the concrete values given in the plots will be added to the revised manuscript.

- **Fig3(c) is not necessary.**

Removed from manuscript as suggested.

- **Fig12 is never discussed in the text.**

The figure is mentioned in p. 15 l 5. However we agree that it needs more clarification in the manuscript.

**References**

C. Bandt and B. Pompe. Permutation entropy: a natural complexity measure for time series. Physical review letters, 88(17):174102, apr 2002. ISSN 0031-9007. doi: 10.1103/PhysRevLett.88.174102. URL http://www.ncbi.nlm.nih.gov/pubmed/12005759.

G. Camps-Valls, J. MunËIJ oz-Mar′Äś, L. Go′ mez-Chova, L. Guanter, and X. Calbet. Nonlinear statistical retrieval of atmospheric profiles from MetOp-IASI and MTG-IRS infrared sounding data. IEEE Transactions on Geoscience and Remote Sensing, 50(5 PART 2):1759–1769, 2012. ISSN 01962892. doi: 10.1109/TGRS.2011.2168963.

C.-C. Chang and C.-J. Lin. Libsvm. ACM Transactions on Intelligent Systems and Technology, 2(3):1–27, 2011. ISSN 21576904. doi: 10.1145/1961189.1961199. URL http://dl.acm.org/citation.cfm?doid=1961189.1961199.

M. Lazaro-Gredilla, M. K. Titsias, J. Verrelst, and G. Camps-Valls. Re- trieval of Biophysical Parameters With Heteroscedastic Gaussian Pro- cesses. IEEE Geoscience and Remote Sensing Letters, 11(4):838–842, apr 2014. ISSN 1545-598X. doi: 10.1109/LGRS.2013.2279695. URL http://ieeexplore.ieee.org/document/6595574/.

M. D. Mahecha, M. Reichstein, N. Carvalhais, G. Lasslop, H. Lange, S. I. Seneviratne, R. Vargas, C. Ammann, M. A. Arain, A. Cescatti, I. a. Janssens, M. Migliavacca, L. Montagnani, and A. D. Richardson. Global convergence in the temperature sensitivity of respiration at ecosystem level. Science (New York, N.Y.), 329(5993):838–40, aug 2010. ISSN 1095-9203. doi: 10.1126/science.1189587. URL http://www.ncbi.nlm.nih.gov/pubmed/20603495.

G. Tramontana, M. Jung, C. R. Schwalm, K. Ichii, G. Camps-Valls, B. Ra′duly, M. Reichstein, M. A. Arain, A. Cescatti, G. Kiely, L. Merbold, P. Serrano-Ortiz, S. Sickert, S. Wolf, and D. Papale. Predicting carbon dioxide and energy fluxes across global FLUXNET sites with regression algorithms. Biogeosciences, 13 (14):4291–4313, jul 2016. ISSN 1726-4189. doi: 10.5194/bg-13-4291-2016. URL http://www.biogeosciences.net/13/4291/2016/.

M. Zanin, L. Zunino, O. A. Rosso, and D. Papo. Permutation Entropy and Its Main Biomedical and Econophysics Applications: A Review. Entropy, 14 (12):1553–1577, aug 2012. ISSN 1099-4300. doi: 10.3390/e14081553. URL http://www.mdpi.com/1099-4300/14/8/1553/.

---

## Author Comment (AC2) · 1 Mar 2017

**Response to Reviewer 2**

In the following, we denote comments by the reviewer in **bold** and our own reponses in standard fonts.

**Review of "Reverse engineering model structures for soil and ecosystem respiration: the potential of gene expression programming"**

We would like to thank the reviewer for the evaluation and detailed comments on our manuscript. We further provide responses for the posed questions and details on how we intend to revise the manuscript.

In this manuscript Ilie et al. explore the use of "gene expression programming" (GEP) to select empirical models for soil and ecosystem respiration. The authors make a case that GEP is a technique for reverse engineering model structures by elucidating underlying mechanisms, rather than depending on hypothesis-driven experiments to identify these mechanisms.

- Indeed, this is our main motivation. But clearly also other methods for reverse engineering may be usable.

I have several concerns about the conceptual framework the authors used to present GEP. I am convinced that GEP is an interesting and worthwhile approach to automate model selection. However, I think it is over-reaching to suggest that GEP can 'reverse engineer' model development. It seems to me that the value of GEP is simply to automate the process of exploring a large number of regression models. I am not convinced that GEP reorganizes the model development process, because regression already is often the first step in model development.

- Thank you for challenging our fundamental ideas. The motivation of this work was indeed to automatize model development. And we believe that a GEP type of approach can help in such an endeavour. But we also agree that GEP is basically doing a selection after rejecting a large number of potential regression models. And this is still very different from classical model building. The choice of the regression model structure is not made directly by the analyst and rather by the algorithm. The analyst comes into play for deciding if and which solution proposed by GEP should be further used. The points discussed here were added to the revised manuscript. p3 l18-20.

Further, I find that the claim that GEP minimizes human influence and perception bias to be strong, as the authors seemingly arbitrarily select the driving

**variables for the model, regardless of how the model's functional form is derived. From other work we know that selecting a single soil temperature at 5 cm soil depth can give a very different model from selecting a temperature from 15 cm soil depth (Graf et al., Biogeosciences doi:10.5194/bg-5-1175-2008). Similarly selecting to use VWC rather than a parameter like matric potential could be the difference between being able to predict rapid increases in flux with rainfall and not.**

- We have provided an initial series of candidate predictors among and GEP automatically does a feature selection. Hence the model development remains a more objective approach. Moreover, GEP is meant to select not only the driver but also the model. Therefore, GEP should be able to deal with cases as the one suggested by the reviewer: different $T_{soil}$ measurement depth can lead to different models. And this was clearly illustrated in the analysis with artificial data.

**In the end, the functions selected by GEP suffer from the same problems as previously used formulae shown in Table 2. All of these functions tend to underestimate large fluxes ("hot spots" and "hot moments"). While the form of the functions may hold-up from training datasets to prediction datasets, the specific parameterizations often do not. I believe the authors have done a good job discussing limitations of GEP, and empirical approaches in general, in section 5.1.1. We know biogeochemical fluxes integrate multiple pools, reservoir dynamics and lags, and these are difficult to detect using semi-empirical models. The largest gains recently in representing soil respiration have come from simulating enzyme kinetics and solute diffusion (e.g. DAMM model) as well as simulating microbial growth dynamics. These advances have come from implementing expert knowledge, not from expediting regression model selection.**

- We agree that we cannot show yet or beat expert knowledge as encoded e.g. in the DAMM model. Still, we believe that our paper is a first step in this direction.

And therefore it is important to showcase this opportunity to the relevant scientific community. The field of reverse engineering is young and cannot look back to half a century of experimental and conceptual work aiming at understanding soil respiration modelling.

**Overall I would recommend that this manuscript be rejected in the current form, and the authors re-evaluate the presentation of the GEP method both in terms of creating certainty within the biogeosciences community that the approach is effective and accessible, as well as readily applicable to field data as was demonstrated with the data from Alice Holt.**

- We do believe that our model approach is readily applicable and a novel tool offering the same accuracy as classical semi-empirical models but crucial with new opportunities of interpretation.

**As was mentioned, I believe the GEP method has considerable potential, but as the manuscript is currently written my concern is that it will pass unnoticed by the community as a whole due to poor accessibility rather than scientific merit.**

- We disagree with this comment, aligning with the other reviewer and also with the overall statement of the strong potential of this novel approach. However, the important step is to get this approach integrated into the modelling community (which is rather small) and allow it to be tested and modified. We do believe that a more general approach and presentation actually will promote its wider usage.

**General Comments:**

1. **I do not agree with Figure 1, that model development starts with expert knowledge. Expert knowledge does not come about on its own, but comes**

[Figure]

**from observations, and regressions are critical to making sense of observations. By helping to identify which variables among a large number of potential explanatory variables correlate to a phenomenon, regression-type analyses lead to the second step in the scientific process: manipulative experiments to confirm hypothesized cause-and-effect relationships. Demonstrating cause-and-effect relationships limits the number of processes that need to be represented in models. I am not convinced that GEP provides a short-cut to this process.**

- We thank the reviewer for his valuable point-of view. Maybe the question is rather what one would call "expert knowledge"? We do see observation as one key element of expert knowledge (Fig 1 now includes " including observations") , leading to a first empirically driven (i.e. regression style) approach to model formulation. Yet, once a model could not be immediately rejected it is propagated and used time and again and refined with including more processes etc. This is a tedious process. And here we see that GEP offers a considerable potential indeed. Maybe we have overstated the value of GEP in the manuscript and we will revise it accordingly, but once again - our motivation was thinking and exploring methods that elegantly bypass this approach. For instance, several of the co-authors have worked on the (Migliavacca et al 2011) paper to build a better model for ecosystem respiration in deciduous forests and come to the conclusion that this should be a job realized by a computer. Figure 1 is changed in the manuscript in order to capture and illustrate the points discussed here as well.

2. **Section 3.1 and 4.1, which outline artificial experiments with the GEP method could be strengthened considerably if the authors were to use a simple, mechanistic model of soil or ecosystem respiration rather than a seemingly random set of algebraic expressions. Using such a respiration**

**model would allow the authors to attempt to recover the model basis functions and, if successful, enhance the reader's confidence with respect to the data from the site at Alice Holt.**

- In this sections we mean to show the capacity of GEP to reconstruct functions from relatively simple example in order to shortly explore the effects of increasing non-linearity and number of variables. As ecological models tend to be more complex and the increase in non-linearity and complexity would no be so clear we chose to stick to some known genetic programming benchmark functions.

  Nevertheless we agree with the reviewer that adding a known ecological respiration model structure in the set of functions to be reconstructed would give more confidence in the application of GEP to ecological modelling. Thus the $Q_{10}$ model is added to the GEP benchmark function set. (p4 l25 and p10 l27-28)

3. **I am concerned about the evaluations of GEP presented in Figs. 3 and 4. Fig 3 compares alternate machine learning techniques by comparing the MEF of the final model selected by each approach. It seems to me also important to compare the actual model structures, not just the fitness score. Did all the techniques recover the original models? If not, is variation in the MEF meaningful?**

   - In this study, GEP is the only approach which gives a readable model structure back. SVM, ANN, RF and KRR lack that property. Thus the comparison is done on the accuracy of predictions, by comparing the modelling scores and residuals.

4. **Figure 4c suggests that GEP was only able to recover about 30-55% of the correct number of parameters. If so, it seems GEP did NOT do a good job of recovering the original models.**

[Figure]

- We agree that at first glance, it would seem bad that the model retrieval with GEP based on the 3 different fitness functions gives a lower number of parameters than the initial number. However considering the high values of MEF when validating against original data, MEF$> 0.96$, we can draw the conclusion that the GEP performed a feature selection, eliminating "low impact" parameters and returned a more simple equivalent solution.

5. **Another major concern is the exercise shown in figure 7. The authors have examined whether summing predicted component fluxes gives predicted total fluxes that resemble observations. This is an interesting idea, but ultimately not that useful for two reasons:**

   (a) **The observed fluxes were not independently measured, e.g. Rauto was not measured independently, but was calculated by measuring the total flux (Rsoil) minus RH. I think you want to test whether all the variability simulated for the components can explain the variability observed for the total flux, but you don't have a measure of the component fluxes independent from the total flux.**

   (b) **We would like to see that the predictions for total flux are no worse than the predictions for the component fluxes. But in several cases the prediction for component fluxes are pretty poor. E.g. Predictions for RECO won't turn out any better than predictions for Rabove, which themselves were poor. That's not so interesting.**

   (a) We agree that because of learning from derived fluxes, it would be hard make a clear statement regarding the capacity of GEP to learn the variability of the studied sum and component fluxes.

   (b) We believe that nevertheless the exercise is useful as it shows that when we use GEP to learn models for each of the flux, sometimes the low-complexity pressure in the fitness functions make that the final solution has a lower

number of parameters and a slightly lower modelling capacity as well. However we see that when we sum up the models of the component fluxes and compare the predictions of these derived models with the original data, although the models have become more complex, the model performance is not significantly improved. This give us more confidence to state that the more simple models retrieved by GEP in the first place have a sufficient capacity to capture the meaningful information present in the data as well.

6. **The manuscript is figure heavy, consider condensing figures or removing. For example can Figures 5 and 9 be combined in an effective way? Are there other figures that may be unnecessary to the reader if they were described in the text or in a table?**

   - Although we agree that the manuscript contains many figures, we believe that they are necessary (or at least helpful) for reflecting the full picture presented in the text.

**Specific comments:**

- **Abstract is long, introduces a lot of terminology. Consider distilling to the most important take-homes, and make more approachable for a general audience.**

  The abstract will be shortened and simplified as suggested.

- **p.3 l. 8. The rationale for reordering should also be to try more options, things that people might miss**

  We would like to thank the reviewer for pointing this out. We agree that the increase in the option pool is a large aspect of our approach and somehow we believed that it would be self-explanatory, however it makes sense to state clearly as well. The aspect is added to the manuscript (p3 l8-9).

- **p. 3. L. 30. Why would we expect the functions to be portable across scales? Provide an ecological justification, otherwise this is not an interesting or useful exercise.**

  We believe that this would be more of a wider discussion of the way in which scaling of ecological models is at all interesting and relevant (Urban 2005).

  What we started exploring here is whether a larger grain model would be capable to capture some very strongly influential divers, even by losing specific information and if such processes indeed appear across scales.

- **p. 3. L. 22-35. When reading initially I found it difficult to understand what hypotheses the authors were testing. I think all of this information is there but needs to be re-organized to make it stand out to the reader.**

  Hypotheses and scope of the paper will be re-organized for clarity as it was suggested by other referee as well.

- **p.4 ll. 5.No need to introduce the conclusions. Consider shortening this to reduce repetition.**

  Thank you for you suggestion. Paragraph removed.

- **2.1 This section was not clearly written, I suggest more careful editing by co-authors. Please avoid including extra words in parantheses, they add complexity without clarity.**

  Section will be re-written for more flow clarity in the revised manuscript as suggested.

- **p.4 ll.15. Is the process of mapping operations to strings relevant to model fitting? I don't think so. Either this is excessive detail about the internal workings of GEP, or you need to explain how this is relevant.**

[Figure]

The process is relevant as it is one of the characteristics of the GEP approach. We apologize for not making this clear in the manuscript already, however this aspect and the effects of mapping are explained in more detail in the method section (2.1) of the revised manuscript.

- **p. 4. L. 20, what do you mean by "solution" The final selected model? Or the respiration predicted by that model? "Genes" and "chromosomes" should be presented in quotations initially.**

  Solution is the final selected model structure. Quotations are added as suggested.

- **p. 4 l. 30 I think you can shorten this paragraph to one sentence, simply state that in each generation, the best variants of a chromosome are determined by a fitness function described below.**

  The paragraph could be shortened, however the suggested line is not accurate as in a generation, there is only a variant for each chromosome , and the fitness function determines the ranking of all chromosomes in that generation.

- **p. 4 l. 32, what is an individual? Do chromosomes make up individuals?**
  An individual is a chromosome that encodes a mathematical formulation, made up by a set of strings called genes.

- **p. 5, l. 1 What is a hyper-parameter? Again, please try to avoid parenthetical phrases in this paragraph.**

  A hyper-parameter is a set of parameters which need to be set for the runs of a certain approach. Definition is added to glossary and further parentheses are avoided.

- **p. 5, ll. 12 "upon request" rather than "on demand".**

  Changed as suggested.

- **p. 5, l. 11-14 most of this information doesn't appear useful, for example, does it actually matter that the cluster had 51 nodes? If someone ran it on a cluster with 12 nodes would it also work but be slower? Either explain the relevance of these details or remove them.**

  The description of the system on which all experiments should be relevant as the results might be influenced by the hardware set-up, due to the initialization of the random seed, speed of solution return and so on. Nevertheless, all non-necessary specification are removed.

- **p. 5, ll. 31 Consider omitting "derived from information-theoretic considerations".**

  Thank you for the suggestion. Omitted.

- **p. 6, ll. 20-25. I didn't understand the reason for this additional optimization. This sounds very much like ordinary regression model selection; does this undermine the unique value of GEP?** The original GEP gives a solution in the form of a general mathematical structure. For accurate scaling a further parameter optimization would be recommended. The value of GEP lays in the capacity of constructing the structure based on the on information found in the input data.

- **p. 6, ll. 27 Scaling noise with signal amplitude: This is good to include! This has been shown for soil respiration too (Lavoie et al. 2015, JGR-Biogeosciences, doi: 10.1002/2014JG002773)**

  Thank you for providing the reference. Added to paragraph.

- **Section 3.2.1 The first two paragraphs are repetitive in describing computation of GPP. Consider omitting or shortening the section on soil flux measurements, since these methods were reported previously.** Would not

remove as they are relevant to experiments and results presented, but will try to shorten.

- **Section 3.2.4 This paragraph can be removed to shorten. Figure 3c, consider omitting. It is repetitive, and the manuscript already has a large number of figures.**

Figure 3c removed. However we believe that the paragraph is needed for anticipating the comparison done on real observation between established models for terrestrial respiration in the community and the GEP based models.

- **p.12, l. 7 Sentence starting "We find that the global modelling performance. . ." Please reword, I don't understand this statement.**

Reworded as suggested.

"We found that when we compared the modelling performance of the models built as sum models from GEP built models for the component fluxes with the original GEP models built on the sum fluxes there not significant differences. However the total number of parameters is much larger for the sum models. This can be a result of the GEP approach eliminating the "low impact" drivers due to complexity pressure. We can see as well that the sensitivity of the sum fluxes to certain drivers can strongly manifest itself only in certain components which is why the drivers only get selected in the models built for those specific components." p12 l7-10.

- **Figure 12, is there a reason that this is presented in a polar plot? It seems on first glance that it could equally be presented as a 4-pane set of cartesian time series plots.**

By using polar plots, we reveal that the seasonal biases of the studied fluxes and the capacity of the models to capture/or not some of the variations in specific times of the year. But yes, it is a matter of taste as well.

**References**

M. Migliavacca, M. Reichstein, A. D. Richardson, R. Colombo, M. a. Sutton, G. Lasslop, E. Tomelleri, G. Wohlfahrt, N. Carvalhais, A. Cescatti, M. D. Ma- hecha, L. Montagnani, D. Papale, S. Zaehle, A. Arain, A. Arneth, T. A. Black, A. Carrara, S. Dore, D. Gianelle, C. Helfter, D. Hollinger, W. L. Kutsch, P. M. Lafleur, Y. Nouvellon, C. Rebmann, R. Humberto, M. Rodeghiero, O. Roup- sard, M. T. Sebastia', G. Seufert, J. F. Soussana, and K. Michiel. Semiem- pirical modeling of abiotic and biotic factors controlling ecosystem respira- tion across eddy covariance sites. Global Change Biology, 17(1):390–409, jan 2011. ISSN 13541013. doi: 10.1111/j.1365-2486.2010.02243.x. URL http://doi.wiley.com/10.1111/j.1365-2486.2010.02243.x.

D. L. Urban. Modeling ecological processes across scales. Ecology, 86(8): 1996–2006, aug 2005. ISSN 0012-9658. doi: 10.1890/04-0918. URL http://doi.wiley.com/10.1890/04-0918.

---

## Referee Report (RR1)

**Comments to the authors**

Geoscientific Model Development – Discussions
Manuscript ID: gmd-2016-242

**Title: Reverse engineering model structures for soil and ecosystem respiration: the potential of gene expression programming**

**Authors: Iulia Ilie, Peter Dittrich, Nuno Carvalhais, Martin Jung, Andreas Heinemeyer, Mirco Migliavacca, James I. L. Morison, Sebastian Sippel, Jens-Arne Subke, Matthew Wilkinson, and Miguel D. Mahecha**

**Major comments**

Overall, the quality of the manuscript has improved. The authors addressed most of my concerns. There is still one point missing, however, that I think is important. It concerns that back transformation of the values (p. 12 l. 20–23). In regression, the correct back transformation of $\log y = x\beta + \varepsilon$ is not $e^{x\beta}$ but $e^{x\beta}E(e^{\varepsilon})$, $E$ being the expectation (e.g. Manning, 1998). For a lognormal case (i.e. the residuals in a the regression of the log-transformed variable are normally distributed), one would get $E(y|x) = e^{x\beta + 0.5\sigma^2}$. Thus, if one transforms without taking the variance of the residuals into account, the transformed values could be biased. This might affect the MEF part of your fitness function. Please explain how you handled this possible bias.

Also, transformation for Random Forest and SVM are rather unusual. While it might somehow help for SVM if you use a Gaussian (i.e symmetric) kernel, a monotonic transformation for decision trees shouldn't change anything.

I understand that the focus of your work is on GEP. However, you compare with other machine learning algorithms and they have their advantages. And one of the advantages of using Random Forest or SVM, for example, is that they do not assume any normality and thus no transformation is necessary. This advantage should be discussed.

I suggest that the authors take care of the issue of back transformation and either revise their calculation or explain why the back transformation in their case is simply the exponential.

**Detailed comments**

- Last point in highlights: either introduce GEP in point 2 or avoid the acronym

- p. 5 l. 1,2 What are a set number of genes and a set fixed length? Do you mean fixed/determined? This is confusing as you also use set in the sense of a mathematical set.

- p. 6 l. 1–2 I don't understand this sentence. Is the word 'one' too much?

- p. 6 l. 3 Is the word 'and' too much?

- p. 7 l. 8 What is the hyper-parameter in GEP? The stopping criterion described above? Why does it have components? Does it mean, there are several hyper-parameters?

- p. 7 l. 13 Replace signature by pattern.

- p. 6 l. 17–19 This sentence is broken.

- p. 7 l. 20–27, 28–31, p. 8 l. 1–8 These paragraphs are repetitive. Try to join the description of the Bandt and Pompe approach.

- p. 8 l. 9 . . . of the fitness . . .

- p. 8 l. 14 number

- p. 8 l. 19–20 Which function is minimized? CEM? MEF+NP? Both? On p. 10 l. 17–19 You state that different functions were used for one experiment only. Which function did you use otherwise?

- p. 9 l. 25 Grammar: "each functional values"

- p. 10 l. 25 The MATLAB toolboxes.

- p. 10 l. 26 "SimpleR" or "Simple R"

- p. 10 l. 25–29 For a reader not familiar with those toolboxes, it is not clear whether you tuned the hyper-parameters. It is not enough to cite the tools. Please state clearly whether you tuned the parameters via a cross-validation.

- p. 11 l. 12 can be calculated

- Section 3.2.3 Please add that you use daily mean values.

- p. 14 l. 28 to p. 15 l. 4 I don't understand the point about summing the predictions. Did you also train the GEP models on summed fluxes? Then, this should be mentioned in the material and method section.

- p. 14 l. 31 How did you test for significance?

- p. 15 l. 12–13 Why is it important that there are nor linear correlations between residuals and predictors? You measure randomness of residuals via the permutation entropy.

- Figure 4 in Supplemental Materials is difficult to read. Please consider splitting it.

**References**

Manning WG. 1998. The logged dependent variable, heteroscedasticity, and the retransformation problem. *Journal of health economics*, **17**(3): 283–295.

---

## Author Response (AR2)

**Overall, the quality of the manuscript has improved. The authors addressed most of my concerns. There is still one point missing, however, that I think is important. It concerns that back transformation of the values (p. 12 l. 20-23). In regression, the correct back transformation of** $logy = x\beta + \epsilon$ **is not** $e^{x\beta}$**, but** $e^{x\beta}E(e^\epsilon)$**,** $E$ **being the expectation (e.g. Manning, 1998). For a log normal case (i.e. the residuals in a the regression of the log-transformed variable are normally distributed), one would get** $E(y|x) = e^{x\beta+0.5\sigma^2}$**. Thus, if one transforms without taking the variance of the residuals into account, the transformed values could be biased. This might affect the MEF part of your fitness function. Please explain how you handled this possible bias.**

**I suggest that the authors take care of the issue of back transformation and either revise their calculation or explain why the back transformation in their case is simply the exponential.**

We would like to thank the reviewer for pointing out the issue regarding the simple exponential back-transformation of regressions trained on logarithmic transformed targets, as we were not aware of the bias which might be introduced in the residuals. Especially since such bias might be stronger when the initial residuals would not be normally distributed. Since this was the case also in our work, we have considered the work presented in the paper referenced by the reviewer and that of others, and we have done a bias correction for all the models and predictions trained on log-transformed targets by multiplying with the exponential of the expectation of the initial residuals as suggested for heteroscedastic residuals. The recalculations for bias-correction were done for all the MLM predictions. As such, the GEP model formulations and the figures and tables referencing the models and their predictions have also been corrected and updated. This was the case for Tables 3 and 4, and figures 6, 7,9,10,11,12,12,14 and 2,3 and 4 of supplement.

We have also added figures to show the effect of bias correction by different approaches as well as plots representing time series of the initial residuals, before back-transformations.

**Also, transformation for Random Forest and SVM are rather unusual. While it might somehow help for SVM if you use a Gaussian (i.e symmetric) kernel, a monotonic transformation for decision trees shouldn't change anything. I understand that the focus of your work is on GEP. However, you compare with other machine learning algorithms and they have their advantages. And one of the advantages of using Random Forest or SVM, for example, is that they do not assume any normality and thus no transformation is necessary. This advantage should be discussed.**

We agree that the log-transformation of a target that has a skewed distribution is especially needed in the case of regression type of modelling , as is the case for GEP, and would not be necessary for the other MLM presented in this paper. Nevertheless, here, we wanted to make a fair comparison and train on the exact same data for a clear picture. Still, since log-transforming is an

extra-step which might not be necessary for the other MLM, we have added this point to the discussion p17 l26-35.

- **Last point in highlights: either introduce GEP in point 2 or avoid the acronym**

  Gene expression programming was used instead.

- **p. 5 l. 1,2 What are a set number of genes and a set length? Do you mean fixed/determined? This is confusing as you also use set in the sense of a mathematical set.**

  We meant set as in fixed/predetermined. Corrected in the manuscript. We apologize for the confusion.

- **p. 6 l. 1-2 I don't understand this sentence. Is the word 'one' too much?** The word 'one' was a leftover from editing indeed. Corrected now.

- **p. 6 l. 3 Is the word 'and' too much?**

  The word 'and' was a leftover from editing indeed. Corrected now.

- **p. 7 l. 8 What is the hyper-parameter in GEP? The stopping criterion described above? Why does it have components? Does it mean, there are several hyper-parameters?**

  The hyper-parameter in GEP is the set of all parameters which need to be fixed before a run of GEP. These are given in Table 1. There is only one hyper-parameter. Clarified in the manuscript as well. p. 6 l. 8

- **p. 7 l. 13 Replace signature by pattern.**

  Replaced as advised.

- **p. 6 l. 17-19 This sentence is broken.**

  Paragraph fixed.

- **p. 7 l. 20-27, 28-31, p. 8 l. 1-8 These paragraphs are repetitive. Try to join the description of the Bandt and Pompe approach.**

  Paragraphs joined.

- **p. 8 l. 9 . . . of the fitness . . .**

  fixed

- **p. 8 l. 14 number**

  fixed

- **p. 8 l. 19-20 Which function is minimized? CEM? MEF+NP? Both? On p. 10 l. 17–9 You state that different functions were used for one experiment only. Which function did you use otherwise?**

CEM was specified as the fitness fucntion used for all experiemnts except the one presented at p. 10 l. 17-19.

- **p. 9 l. 25 Grammar: each functional values"**

  Corrected to "..all functional transformations we done on"...

- **p. 10 l. 25 The MATLAB toolboxes.**

  Corrected

- **p. 10 l. 26 SimpleR" or Simple R"**

  Corrected.

- **p. 10 l. 25-29 For a reader not familiar with those toolboxes, it is not clear whether you tuned the hyper-parameters. It is not enough to cite the tools. Please state clearly whether you tuned the parameters via a cross-validation.**

  Added specifications regarding the estimation of the hyper-parameters.

- **p. 11 l. 12 can be calculated**

  Corrected.

- **Section 3.2.3 Please add that you use daily mean values.**

  Added.

- **p. 14 l. 28 to p. 15 l. 4 I don't understand the point about summing the predictions. Did you also train the GEP models on summed fluxes? Then, this should be mentioned in the material and method section.**

  We did not train again on the summed fluxes, but we only compared the summed predictions of component fluxes with the observations of the sum flux. With this exercise we wanted to see if the summed predictions of component fluxes are capable to reconstruct a sum flux not seen during training, and we believe that the fact that they can confirms that the respiration type separations made sense from a mechanistic point of view.

- **p. 14 l. 31 How did you test for significance?**

  By using the t-test. Added to manuscript.

- **p. 15 l. 12-13 Why is it important that there are nor linear correlations between residuals and predictors? You measure randomness of residuals via the permutation entropy.**

  The lack of correlation of residuals to predictors can be indeed a measure of randomness, but we wanted to see as well if there was any strong relation of predictors in the residuals that was missed during the learning process of the GEP. Since this was not the case, it might mean that the patterns which are still present in the residuals cannot be attributed to only one predictor and are more complex.

- **Figure 4 in Supplemental Materials is difficult to read. Please consider splitting it.**

  Figure was split.

[revised manuscript text omitted]